# Transcription factor family-specific DNA shape readout revealed by quantitative specificity models

Lin Yang[1,†], Yaron Orenstein[2,†,‡], Arttu Jolma[3], Yimeng Yin[3], Jussi Taipale[3], Ron Shamir[2,*] &
Remo Rohs[1,**]

## Abstract

Transcription factors (TFs) achieve DNA-binding specificity through contacts with functional groups of bases (base readout) and readout of structural properties of the double helix (shape readout). Currently, it remains unclear whether DNA shape readout is utilized by only a few selected TF families, or whether this mechanism is used extensively by most TF families. We resequenced data from previously published HT-SELEX experiments, the most extensive mammalian TF–DNA binding data available to date. Using these data, we demonstrated the contributions of DNA shape readout across diverse TF families and its importance in core motif-flanking regions. Statistical machine-learning models combined with feature-selection techniques helped to reveal the nucleotide position-dependent DNA shape readout in TF-binding sites and the TF family-specific position dependence. Based on these results, we proposed novel DNA shape logos to visualize the DNA shape preferences of TFs. Overall, this work suggests a way of obtaining mechanistic insights into TF–DNA binding without relying on experimentally solved all-atom structures.

**Keywords** binding specificity; DNA shape; feature selection; quantitative modeling; transcription factor
**Subject Categories** Genome-Scale & Integrative Biology; Structural Biology; Transcription
**Mol Syst Biol. (2017) 13: 910**

## Introduction

Protein–DNA interactions play a central role in gene regulation. Transcription factors (TFs) are proteins that recognize specific DNA sequences. They bind to regulatory regions in the genome and consequently activate or repress transcription of target genes. TFs can bind various DNA sequences with different DNA-binding affinities or specificities. In the last decade, technologies for measuring protein DNA-binding specificities have advanced tremendously (Slattery *et al*, 2014). Platforms based on microarray technology, such as protein-binding microarray (PBM; Berger *et al*, 2006), and high-throughput sequencing technology, such as high-throughput SELEX (HT-SELEX; Jolma *et al*, 2010) or SELEX-seq (Slattery *et al*, 2011), have enabled measurements of protein binding against thousands or even millions of different DNA sequences. The computational challenges are to develop accurate and quantitative models of protein–DNA binding specificities from these massive datasets and to infer binding mechanisms.

Position weight matrix (PWM) or PWM-like models are widely used to represent DNA-binding preferences of proteins (Stormo, 2000). In these models, a matrix is used to represent the TF-binding site (TFBS), with each element representing the contribution to the overall binding affinity from a nucleotide at the corresponding position. An inherent assumption of traditional PWM models is position independence; that is, the contribution of different nucleotide positions within a TFBS to the overall binding affinity is assumed to be additive. Although this approximation is broadly valid, nevertheless, it does not hold for several proteins (Man & Stormo, 2001; Bulyk *et al*, 2002). To improve quantitative modeling, PWM models have been extended to include additional parameters, such as *k*-mer features, to account for position dependencies within TFBSs (Zhao *et al*, 2012; Mathelier & Wasserman, 2013; Mordelet *et al*, 2013; Weirauch *et al*, 2013; Riley *et al*, 2015). Interdependencies between nucleotide positions have a structural origin. For example, stacking interactions between adjacent base pairs form the local three-dimensional DNA structure. TFs have preferences for sequence-dependent DNA conformation, which we call DNA shape readout (Rohs *et al*, 2009, 2010).

Based on this rationale, an alternative approach to augment traditional PWM models is the inclusion of DNA structural features. Models of TF–DNA binding specificity incorporating these DNA shape features achieved comparable performance levels to models

1  Molecular and Computational Biology Program, Departments of Biological Sciences, Chemistry, Physics & Astronomy, and Computer Science, University of Southern California, Los Angeles, CA, USA
2  Blavatnik School of Computer Science, Tel Aviv University, Tel Aviv, Israel
3  Division of Functional Genomics and Systems Biology, Department of Medical Biochemistry and Biophysics, Karolinska Institutet, Stockholm, Sweden
   *Corresponding author. Tel: +972 3 640 5383; E-mail: rshamir@tau.ac.il
   **Corresponding author. Tel: +1 213 740 0552; E-mail: rohs@usc.edu
   † These authors contributed equally to this work
   ‡ Present address: Computer Science and Artificial Intelligence Laboratory, Massachusetts Institute of Technology, Cambridge, MA, USA

incorporating higher-order *k*-mer features, while requiring a much smaller number of parameters (Zhou *et al*, 2015). We previously revealed the importance of DNA shape readout for members of the basic helix-loop-helix (bHLH) and homeodomain TF families (Dror *et al*, 2014; Yang *et al*, 2014; Zhou *et al*, 2015). We were also able, for Hox TFs, to identify which regions in the TFBSs used DNA shape readout, demonstrating the power of the approach to reveal mechanistic insights into TF–DNA recognition (Abe *et al*, 2015). This capability was extensively shown for only two protein families, due to the lack of large-scale high-quality TF–DNA binding data. With the recent abundance of high-throughput measurements of protein–DNA binding, it is now possible to dissect the role of DNA shape readout for many TF families.

In this study, we used the most extensive mammalian TF–DNA binding affinity datasets available to date, derived from HT-SELEX experiments (Jolma *et al*, 2013), to inform DNA shape-based binding models. To improve statistical robustness of the analysis, we augmented each experiment by increasing the sequencing depth of existing HT-SELEX data (Jolma *et al*, 2013). We implemented a pipeline to derive accurate TF-binding intensities for all possible DNA *M*-words (sequences of length *M*) from HT-SELEX reads. Using these preprocessed data, we trained machine-learning models of TF–DNA binding specificities. Finally, using feature selection, we pinpointed positions in the TFBSs where DNA shape readout is most likely to occur.

## Results

### HT-SELEX experimental data provide accurate *M*-word scores for diverse TF families

We analyzed HT-SELEX data, including 548 experiments covering 410 human and mouse proteins from 40 different TF families, to produce *M*-word binding scores. Increased sequencing depth allowed us to derive accurate scores for longer *M*-words. This aspect is particularly important because DNA shape is affected by the flanking regions of TFBSs. Therefore, we augmented the original dataset (Jolma *et al*, 2013) with additional sequencing to increase the read depth of the experiments by almost 10-fold (from an average of ~168,000 reads per sequencing file to ~1,656,000 reads). Experimental data were filtered by rigorous quality control (QC) criteria to identify cases with sufficient library complexity and read counts to allow the building of multiparametric models. A total of 218 TFs from 29 families passed the first filter based on high variability and large sample size of the data, and a total of 215 TFs from 27 different families passed the QC step based on regression performance (Fig 1).

For each TF, we selected a core-binding motif, to enable identification of the most probable binding site within *M*-words and filter out oligonucleotides that are likely to be unbound. The motifs used were derived from a previous study (Jolma *et al*, 2013). These motifs generally contain long flanks in addition to the core consensus sequence, which would prevent us from getting robust *M*-word scores due to low read coverage for long sequences. To overcome this difficulty, we used motifs from the catalogue compiled by Weirauch and Hughes (Weirauch & Hughes, 2011) to identify and use only the core positions. We calculated the binding score for each *M*-word that included the core motif in the center (allowing for a

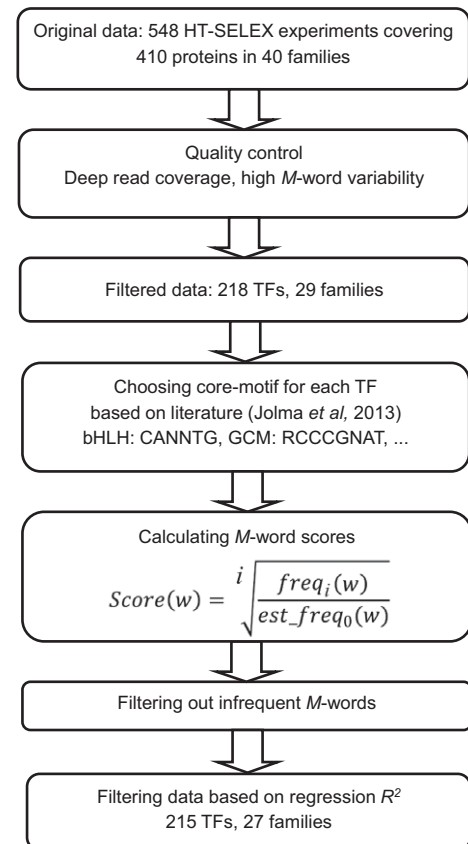

**Figure 1.  Pipeline used to generate HT-SELEX *M*-word scores and filter datasets.**
*M*-word scores were derived for cycles $i \geq 3$. For the calculation of the scores, $freq_i(w)$ is the frequency of *M*-word $w$ in cycle $i$, and $est\_freq_0(w)$ is its estimated frequency in cycle *0*.

few mismatches) and any possible flanking sequences 5′ and 3′ of the motif. We sought to avoid the possibility of cooperative TF–DNA binding, in which multiple copies of the TF occupy different DNA-binding sites (BSs) on the same sequence, as well as to minimize noise caused by inaccurate alignment of *M*-words based on the core motif. Thus, we excluded HT-SELEX reads that contained multiple instances of the core motifs.

Next, we derived *M*-word binding scores based on observed experimental enrichment. Each HT-SELEX experiment included several rounds of binding site (BS) selection by the TF, with the binding specificity of selected DNA sequences increasing in each round. We calculated the *M*-word score as the ratio of the frequency of the *M*-word in round *i* over its estimated frequency in the initial round, using a fifth-order Markov model (Slattery *et al*, 2011). The final output of this process was the *M*-word scores of the core sequence and its flanks for each HT-SELEX experiment (Appendix Fig S1A).

To evaluate the accuracy of our *M*-word scoring scheme and the value of deeper sequencing, we compared scores derived by HT-SELEX to those measured by genomic-context PBMs (gcPBMs). The gcPBMs use arrays specifically designed with the core sequence in the center, flanked by a genomic context (Gordân *et al*, 2013). These probes are intended to measure the effect of flanking sequences and, therefore, provide an accurate gold standard for

long *M*-word ($M \geq 12$) binding scores. The only protein for which both gcPBM and HT-SELEX experimental data exist was the Max homodimer (Zhou *et al*, 2015). Appendix Fig S1B shows the good correlation ($r = 0.64$) of 12-word scores produced by the two technologies, demonstrating the accuracy of our process in producing *M*-word scores from HT-SELEX data. To test how much we gain with respect to gcPBM binding scores by using the new data, we examined three different *M*-word scores: frequency, ratio compared with the initial round, and ratio compared with the estimated initial round. Deeper sequencing improved the correlation of these three scores to gcPBM 12-word scores, and the ratio-to-estimated score achieved the highest correlation (Appendix Fig S1C). Notably, when processing the data previously published in (Jolma *et al*, 2013) with the same pipeline, only 22 proteins passed the quality control, compared with 218 with the higher coverage, showing the advantage of deeper sequencing.

## Principal component analysis (PCA) reveals TF family-specific DNA-binding specificities and heterogeneities within TF families

We performed PCA to visualize TF family-specific DNA-binding specificities. The DNA-binding preference of each TF was represented by the DNA *M*-word with the highest binding affinity for this TF. We encoded this *M*-word into numeric feature vectors that included (i) only mononucleotide (i.e., 1-mer) features, and (ii) both 1-mer and DNA shape features. DNA shape features include minor groove width (MGW), Roll, propeller twist (ProT), and helix twist (HelT) and are predicted with our DNAshape approach (Zhou *et al*, 2013). Figure 2A and B shows the first two principal components obtained using each feature vector.

Different TF families tended to form distinct clusters in the PCA scatter plots. To compare the clustering quality in the two plots, we obtained the two-dimensional Euclidean distances between all pairs of TFs from Fig 2A and B. Distances were classified as intra- or inter-family and visualized as boxplots (Fig 2C and D). Inter-family distances were generally larger than intra-family distances. When we used both 1-mer and DNA shape features, the difference between the medians of the inter- and intra-family groups was slightly larger than the difference obtained when using 1-mer features alone (Fig 2C and D). This result was consistent with Fig 2A and B, indicating that more variance could be explained by introducing DNA shape features, in part due to the better separation of the homeo-domain family (Fig 2B). To test whether such effects were simply due to the higher dimensionality introduced by the additional DNA

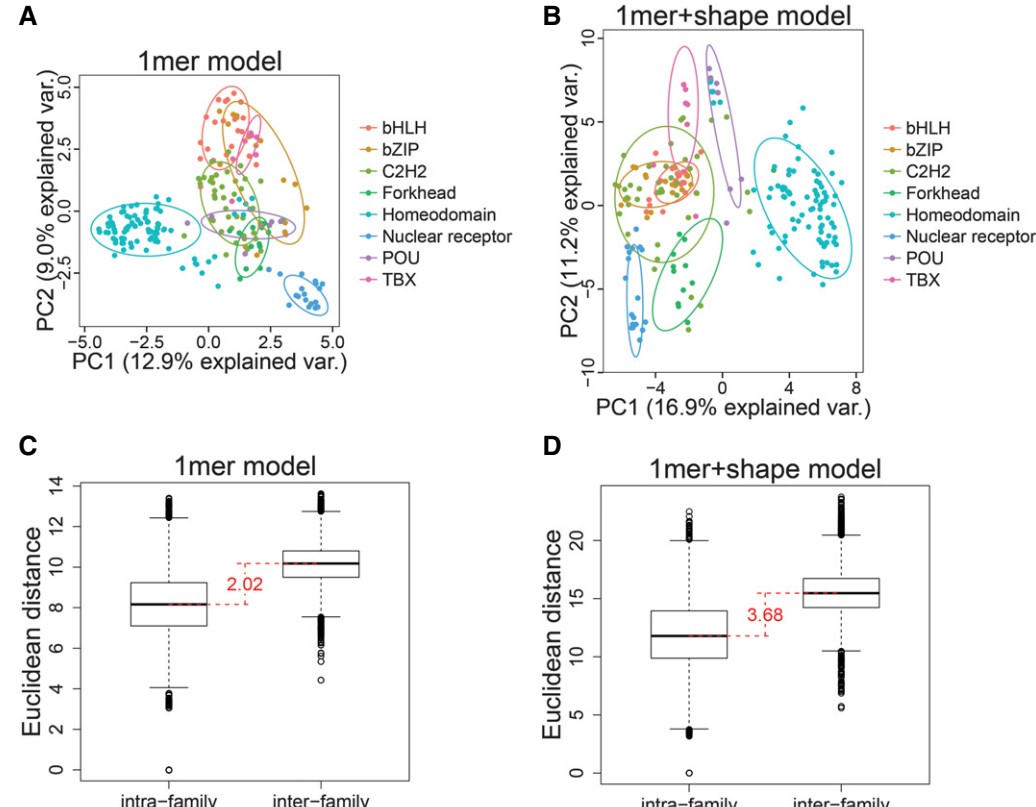

**Figure 2.  PCA reveals different DNA-binding specificities between TF families.**

A   PCA using 1-mer features. Each dot represents a TF. Dots of the same color belong to the same TF family. An ellipse was drawn for each TF family. The ellipse is a contour of a fitted two-variate normal distribution that encloses 0.68 probability (R package default).

B   PCA using 1-mer and shape features, annotated in the same way as described in (A).

C   Boxplots of inter- and intra-family TF distances derived from (A). Difference between medians of inter- and intra-family distances is 2.02 (red).

D   Boxplots of inter- and intra-family TF distances derived from (B). Difference between medians of inter- and intra-family distances is 3.68 (red).

shape features, we added randomly generated shape features based on Gaussian distribution with mean and standard deviation of the original shape features. Both the variance explained and the distance between intra- and inter-family groups were lower in this test (Appendix Fig S2).

## DNA shape features improve modeling of DNA-binding specificities across TF families

We tested the importance of the recognition of DNA shape by each TF through quantitative modeling of DNA-binding specificities and comparison of model performance in terms of the $R^2$ between predicted and experimental *M*-word scores. Similar to the methodology in Yang *et al* (2014) and Zhou *et al* (2015), we built regression models that used only DNA mononucleotide features (i.e., 1mer models) or that combined DNA mononucleotide and shape features (i.e., 1mer+shape models). A result in which the 1mer+shape model outperforms the 1mer model indicates that DNA shape readout might play a role in TF binding.

Based on an analysis of 215 TFs from 27 different families, we found that 1mer+shape models generally outperformed 1mer models (Fig 3A), indicating the prevalence of DNA shape readout across different TF families (for a complete list of datasets used in Fig 3, see Table EV1). With DNA sequence readout playing a dominant role in TF binding, the importance of DNA shape recognition as additional contribution varied both between and within TF families. For example, model performance for homeodomain TFs was generally more substantially improved than for C2H2 TFs. Within the homeodomain TF family, there was a large variance among individual members. Homeodomain and bHLH TFs have been previously observed to be sensitive to DNA shape features (Slattery *et al*, 2011; Gordân *et al*, 2013; Yang *et al*, 2014; Zhou *et al*, 2015). Here, we confirmed and extended this observation to the bZIP, CENPB, CP2, CUT, ETS, HSF, IRF, MYB, NFAT, nuclear receptor, PAX, POU, PROX, TBX, and TEA TF families. At least half of the members in each of these families, covered by our data, showed greater than 10% performance improvement when DNA shape features were added to the model. However, some families were underrepresented in the data with only one TF present (Table EV1; for full names and detailed information of the TF families, see Table EV2).

To test the robustness of the experimental data and our computational pipeline, we repeated the above analysis on replicate experimental data for three TFs from the bHLH and homeodomain families. Our results consistently showed contributions of DNA shape readout for these two families (Appendix Fig S3A). To test whether the performance gain is simply a result of the increased number of model parameters due to the added DNA shape features, we shuffled the query table for DNA shape features. Shape models based on the shuffled query table generally have poorer performance than those based on the original query table (Fig 3B). We also tested whether the results were robust to the motif seeds used during data preprocessing. We repeated the above analyses using the Weirauch and Hughes seeds (Weirauch & Hughes, 2011) as the final seeds instead of using them for identifying the core positions of the HT-SELEX-based motifs published by Jolma *et al* (2013). We calculated Pearson's correlation coefficients between the performance of models that were based on the Weirauch and Hughes seeds (Weirauch & Hughes, 2011) and the Jolma *et al* (2013) seeds.

The high correlation between the two sets of motif seeds indicated that the results were robust to the choice of motif seeds (Appendix Fig S3B). We also tested the robustness of the results under slight changes in the mismatch threshold (see Materials and Methods) and length of the flanking regions. Both tests showed high correlation between different parameter settings, demonstrating sufficient robustness (Appendix Fig S3C and D).

The homeodomain TFs in this study presumably bind DNA as monomers, whereas our previous studies demonstrated the importance of DNA shape for Exd–Hox heterodimers (Slattery *et al*, 2011). X-ray and nuclear magnetic resonance (NMR) structures of homeodomain DNA-binding domains in complex with DNA repeatedly show that the N-terminal tail of the homeodomain DNA-binding domain interacts with the DNA through minor groove and backbone contacts, which is a signature of DNA shape readout (Joshi *et al*, 2007).

## DNA shape features in flanking regions are important for different TF families

We previously observed that 1mer+2mer+3mer models usually outperform 1mer+shape models (Zhou *et al*, 2015). Here, we gained additional clues for possible explanations of this observation. As noted previously (Zhou *et al*, 2015), both 2-mer and 3-mer features are indirect representations of DNA shape characteristics. The 2-mer features describe stacking interactions between adjacent base pairs, whereas 3-mer features describe short structural elements, such as A-tracts that tend to form narrow minor groove regions. Thus, it is not surprising that 1mer+2mer+3mer models can capture TF–DNA binding specificities with high accuracy.

Using our high-quality HT-SELEX data, we observed that, for most TFs, 1mer+2mer+3mer models outperformed 1mer+shape models (Fig 3C). As our prediction of local DNA shape features was based on a sliding window of 5 base pairs (Zhou *et al*, 2013), we were unable to predict shape features for the two extreme positions at the 5′ and 3′ ends of each DNA sequence. This limitation could give an edge to 1mer+2mer+3mer models. However, we could encode 2-mer and 3-mer features for those terminal positions, which in turn would work as a proxy for DNA shape. To test this hypothesis, we added 3-mer features from only the two end (E2) positions (i.e., 3merE2 features) to the 1mer+shape model. Performance of the resulting 1mer+shape+3merE2 model was indeed comparable to that of the 1mer+2mer+3mer model (Fig 3D). As an additional test, we removed 2-mer and 3-mer features at the end positions from the 1mer+2mer+3mer model, which resulted in the 1mer+2merNoE2+3merNoE2 model that showed similar performance to the 1mer+shape model (Fig 3E).

We also hypothesized that if longer flanking sequences were available for predicting shape features, then 1mer+shape models would perform similar to 1mer+2mer+3mer models without adding 3merE2 features. To verify this possibility, we used an independent dataset generated by the gcPBM platform (Zhou *et al*, 2015). As expected, 1mer+shape models performed comparable to 1mer+2mer+3mer models for the data without additional 3merE2 features (Appendix Fig S3E). These results imply that DNA shape features in the flanking regions contribute to TF–DNA binding specificities, which was previously known for bHLH TFs (Gordân *et al*, 2013; Yang *et al*, 2014; Zhou *et al*, 2015). Here, we showed for the first

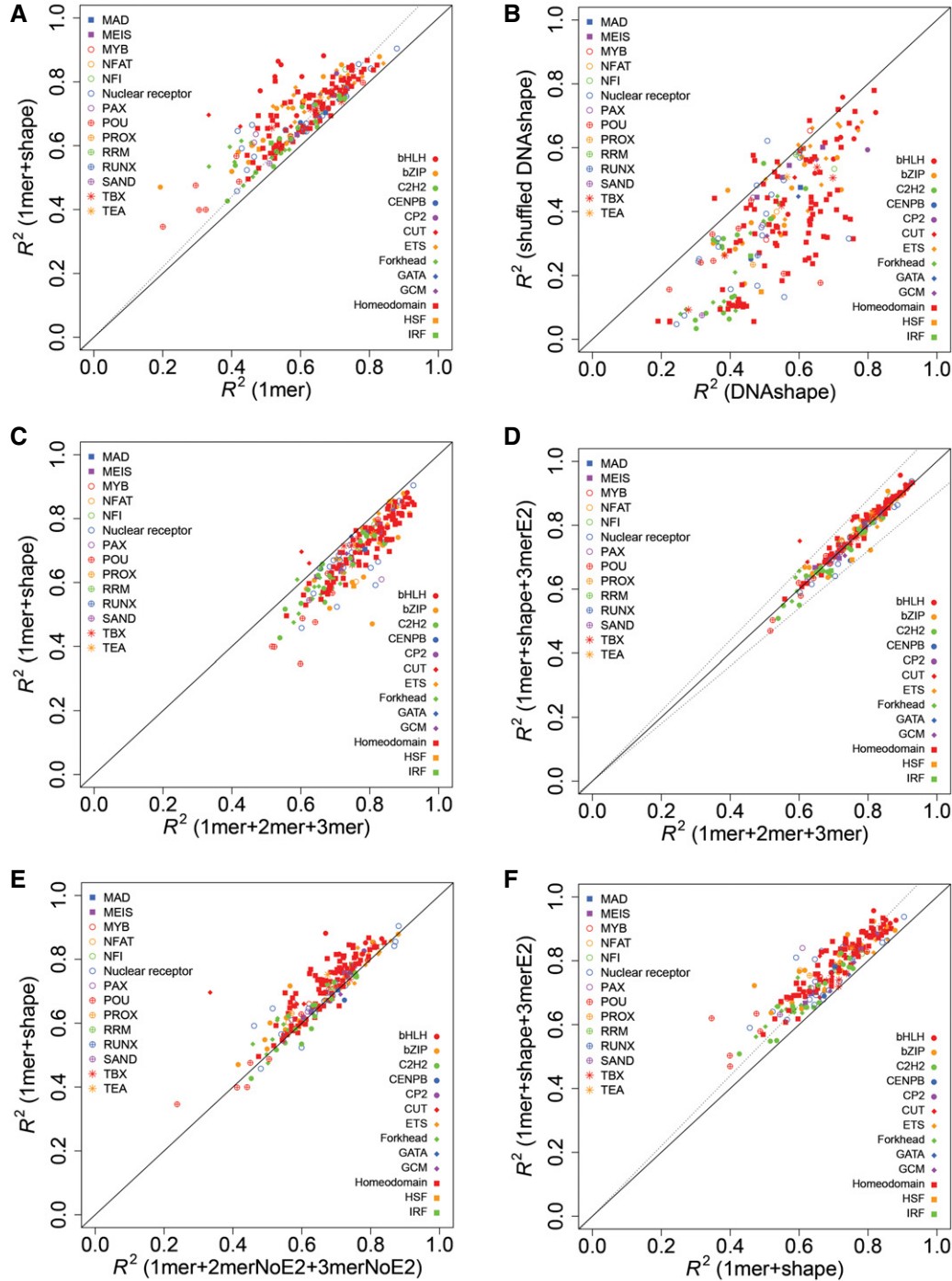

**Figure 3.  Performance comparisons between models using different features.**

A   Comparison between 1mer and 1mer+shape models.

B   Comparison between shape models that are based on the original DNAshape method (Zhou *et al*, 2013) and randomly shuffled pentamer query tables.

C   Comparison between 1mer+2mer+3mer and 1mer+shape models.

D   Comparison between 1mer+2mer+3mer and 1mer+shape+3merE2 models. The label 3merE2 represents 3mer features from the two end positions at the 5' and 3' terminal of each DNA sequence.

E   Comparison between 1mer+2merNoE2+3merNoE2 and 1mer+shape models. The labels 2merNoE2 and 3merNoE3 indicate that 2mer and 3mer features, respectively, were removed from the end positions.

F   Comparison between 1mer+shape and 1mer+shape+3merE2 models.

Data information: Each dot represents one dataset. Coordinates of the dot are determined by the performance, measured in $R^2$ based on 10-fold cross-validation, of the corresponding models indicated in parentheses. Shape and color of the dots indicate the TF family. Dashed lines in (A and F) have a slope of 1.1, indicating 10% performance increase. Dashed lines in (D) have slopes of 1.1 and 0.9.

time that this phenomenon is of general nature, as adding 3merE2 features as proxy for missing DNA shape features consistently improved the model performance for various TF families (Fig 3F).

Beyond better interpretability of shape-augmented models, an important distinction between the models is the different number of features required to achieve similar performance. The 1mer+shape model requires 12 features (including second-order DNA shape features) per nucleotide position compared with the 84 features required by the 1mer+2mer+3mer model per nucleotide position (Zhou *et al*, 2015). Although we previously included lower-order 1-mers and 2-mers in our 1mer+2mer+3mer models for reasons of interpretability, nevertheless, the 3-mer features actually contain all of the information of the 1-mers and 2-mers. Thus, a 3mer model is equivalent to a 1mer+2mer+3mer model (Materials and Methods and Appendix Fig S3F). This choice, however, would still leave the 3mer model with 64 required features per nucleotide position compared with a maximum of only 12 features in the 1mer+shape model.

## Feature selection can provide insights into TF–DNA readout mechanisms

We performed feature selection to identify BS positions where DNA shape features contribute to TF-binding specificities. The method is similar to the one we previously introduced for the analysis of SELEX-seq data for Hox proteins (Abe *et al*, 2015). For each TF, we evaluated the $R^2$ performance of the baseline 1mer model, denoted $R^2_{1mer}$. Next, we evaluated models that combined 1-mer features with DNA shape features individually at single nucleotide positions $i$, denoted 1mer+shape$_i$ models. We denoted the performance as $R^2_{1mer+shape_i}$. We calculated the difference in model performance $\Delta R^2_i = R^2_{1mer+shape_i} - R^2_{1mer}$ for each nucleotide position $i$ (Fig 4A). The $\Delta R^2_i / R^2_{1mer}$ ratio indicates the percentage change in performance due to the availability of DNA shape features at nucleotide position $i$, with a positive ratio suggesting performance gain. The ratio at position $i$ compared with other positions reflects the relative importance of DNA shape features at different nucleotide positions. We visualized the $\Delta R^2_i / R^2_{1mer}$ ratio as a function of position $i$ for each TF in the form of a heat map (Fig 5A and Appendix Fig S4).

To avoid interference from DNA sequence information, we devised a second feature-selection approach in which we removed DNA shape features at individual positions from a shape-only model. The $\Delta R^2_i / R^2_{shape}$ ratio was then used for generating the heat map (Figs 4B and 5B, and Appendix Fig S4), where $\Delta R^2_i = R^2_{shape} - R^2_{shape_i}$. These two different approaches can sometimes yield conflicting heat maps as discussed below. To address such cases and facilitate the use of these heat maps, we also generated a combined heat map based on the cell-by-cell minimum of the two heat maps (Fig 5C and Appendix Fig S4). Quantitative information about the importance of the position-dependent DNA shape in TF–DNA recognition at single-base pair resolution provides the means to determine the structural protein–DNA readout mechanisms based on sequence data. To achieve this goal, we further expanded our feature-selection method to test each individual DNA shape feature category, which enabled us to gauge the importance of each DNA shape feature, that is, MGW, Roll, ProT, or HelT, at every position (Appendix Fig S5). To date, obtaining such information required experimentally solved structures.

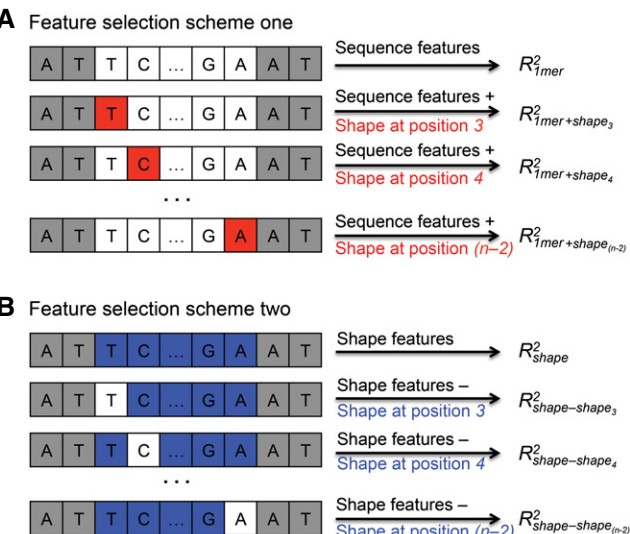

**Figure 4.    Schematic representation of feature-selection process.**

A  Feature-selection scheme for adding DNA shape features at one individual position to a sequence-only model.

B  Feature-selection scheme for removing DNA shape features from one single position from a shape-only model.

Figure 5 shows the position-dependent DNA shape importance for homeodomain TFs that recognize a TAAT motif. For most of these TFs, DNA shape was more important at the 3′ side of the core motif, as indicated by the darkness of colors (Fig 5). Homeodomain TFs that recognize a different motif, for example, TCRTAAA, were shown to have a different positional DNA shape preference (Appendix Fig S4F). Positional preferences were also protein-family specific. For example, for bHLH TFs DNA shape features in both flanking regions were important, whereas for nuclear receptors that bind to an ACANNNTGT motif the central motif region was generally important (Appendix Fig S4A and H). In comparison, bZIP TFs that bind to a TTRCGC motif and homeodomain TFs were generally sensitive to DNA shape features at only one flanking side of the core motif (Appendix Fig S4B and F).

The exact positions where DNA shape features are important were not unambiguously pinpointed for the bHLH TFs and the nuclear receptors that bind to an ACANNNTGT motif (Appendix Fig S4A and H). Both Appendix Fig S4A and H relate to a scenario where the red heat map shows prominent shape effects in multiple consecutive positions, whereas the blue heat map shows almost no effects. We believe that this is due to false positives in the red heat map, that is, positions that are not important for shape readout but identified as such, and false negatives in the blue heat map, that is, positions that are important for shape readout that were not identified. We conclude in this case that DNA shape is important in some positions in the consecutively red regions, but we failed to locate it, even with the help of the blue heat map.

We illustrated the relevance of feature importance heat maps derived from feature-selection approaches by considering experimental structures of the homeodomain proteins PITX2 (PDB ID 2LKX) and GBX1 (PDB ID 2ME6) in complex with DNA (Fig 6A and B). These structures provide possible explanations for entries

representing PITX3 and GBX1 on the heat maps (Fig 5). As no experimental structure for PITX3 is available, we used an NMR structure for PITX2 (Chaney *et al*, 2005), which shares the same

DNA-binding domain as PITX3. In the heat maps, PITX3 has darker colors at the 3′ side of the TAAT motif, indicating a more important role of DNA shape at these positions. In the PITX2

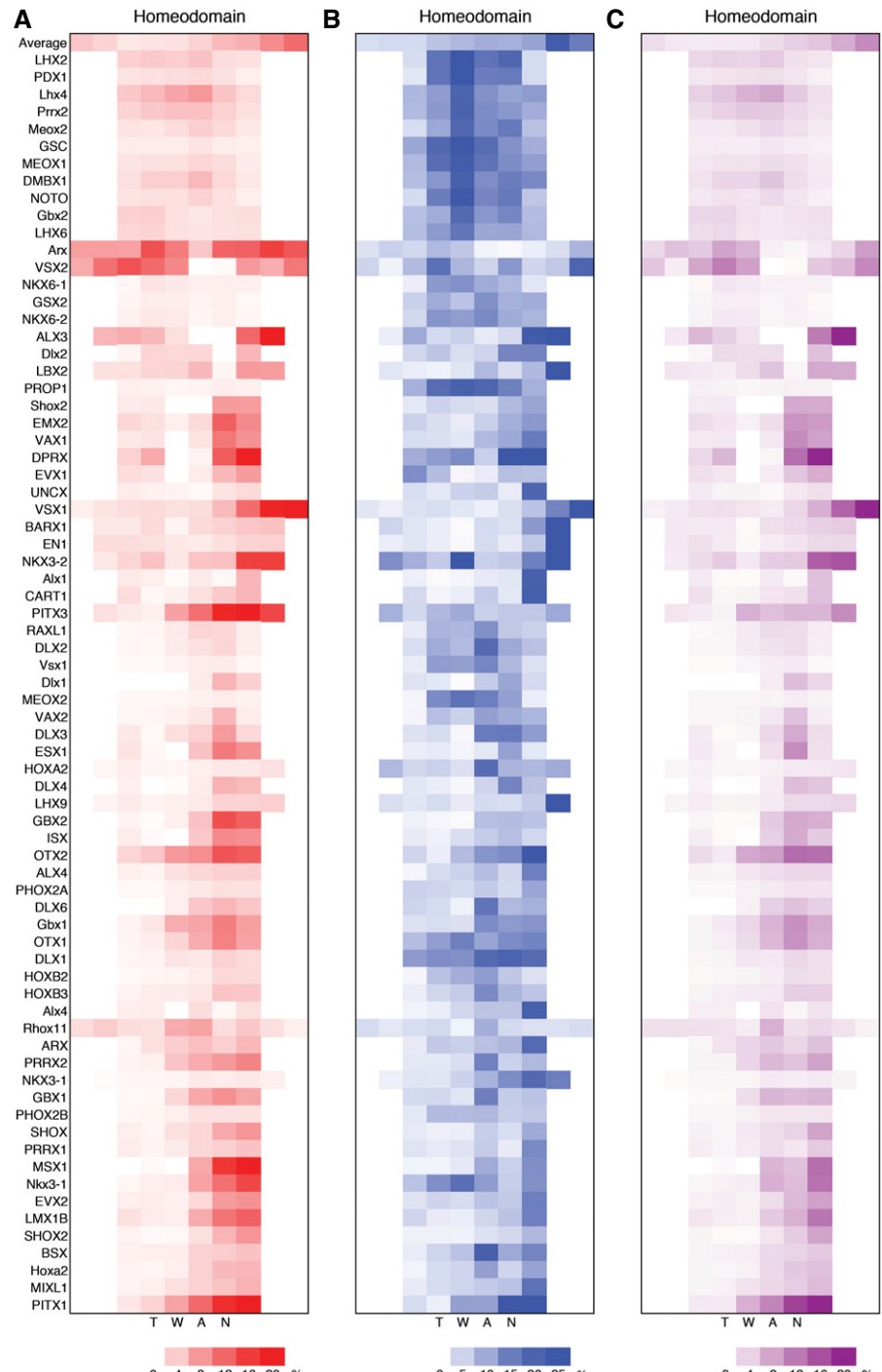

**Figure 5.  Importance of DNA shape features as a function of nucleotide positions revealed by feature selection with machine learning.**

A   Heat map based on adding DNA shape features to a sequence-only model.

B   Heat map based on removing DNA shape features from a shape-only model.

C   Combined heat map that takes cell-by-cell minimum of heat maps in (A and B).

Data information: Case of letters in TF names indicates species, with uppercase being human and lowercase being mouse.

structure, the N-terminal tail of the protein interacts with DNA in the minor groove of the TAAT motif. The structure contains a narrow minor groove region near the second A within the TAAT motif (Fig 6A). In this case, the protein might exploit the DNA structural characteristics at positions highlighted in the heat maps to achieve its binding specificity.

We observed similar concurrence between heat map and structural analyses for the TF GBX1, where the structure has a narrow minor groove region at the 3′ flank (Fig 6B). Although the positions indicated by the heat maps do not match the positions in the structure in an exact way, the heat maps successfully highlighted those nearby positions. Moreover, the heat maps were consistent with our conclusion that DNA shape features in flanking regions are important for TF–DNA binding specificities (Fig 3D–F). In addition to the homeodomain family, we used a structure of the human progesterone receptor (PDB ID 2C7A) from the nuclear receptor family to illustrate how the heat maps can provide hints to the structural mechanisms of protein–DNA

binding. In the structure (Roemer *et al*, 2006), MGW, Roll, and ProT show distinct characteristics in the central region of the DNA-binding site, which potentially explains the central "red" regions in the heat maps (Appendix Fig S6).

### DNA shape logos represent structural readout mechanisms

To visualize the detailed DNA shape preferences of individual TFs, we propose a new visualization, *DNA shape logos*, analogous to sequence logos for PWMs. In these logos, we used the letters H, M, P, and R to represent DNA shape features HelT, MGW, ProT, and Roll, respectively. The height of each letter indicates the importance derived from the feature-selection analysis for the corresponding DNA shape feature at a specific position (Fig 6). As an example, we used $\Delta R^2$, that is, the performance gain due to adding an individual DNA shape feature to a 1mer model, to generate shape logos for PITX3 and GBX1 (Fig 6C and D). For PITX3, a prominent M at positions 7, 8, 9, and 10 overlaps with the narrow minor groove region

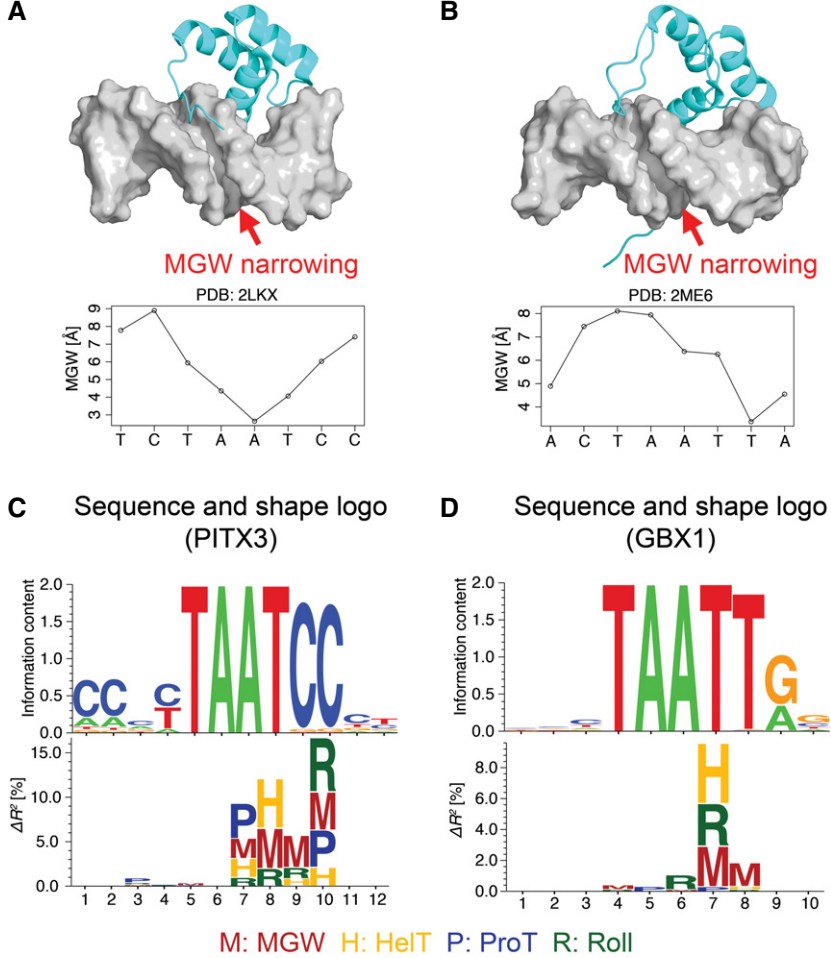

**Figure 6.  Three-dimensional structure and DNA sequence and shape logos for the homeodomain TFs PITX2/PITX3 and GBX1.**

A    NMR structure of PITX2 in complex with DNA (PDB ID 2LKX) and the CURVES (Lavery & Sklenar, 1989) derived plot for the MGW of the bound DNA.
B    NMR structure of GBX1 in complex with DNA (PDB ID 2ME6) and the CURVES (Lavery & Sklenar, 1989) derived plot for the MGW of the bound DNA.
C    DNA sequence and shape logos for PITX3.
D    DNA sequence and shape logos for GBX1.

in the structure. Similarly, for GBX1, a prominent M at positions 7 and 8 overlaps with the narrow minor groove in the structure. DNA shape information was missing for the two nucleotide positions at each end of the TFBS; thus, no letters are shown at these positions in the shape logo. DNA shape logos can facilitate the integration of structural information in motif finding tools. Sequence and shape logos for all the TFs studied in this work are provided as Datasets EV1 and EV2, respectively.

## Discussion

Protein–DNA binding models have evolved tremendously in the last decade (Slattery *et al*, 2014). In the past, binding models were based on a few high-affinity BSs. These models enabled the identification and prediction of the most likely BSs *in vivo*, but missed many potential low-affinity sites (Stormo, 2000; Tanay, 2006). Weak and suboptimal TFBSs play important roles in transcriptional regulation (Crocker *et al*, 2015; Farley *et al*, 2015), emphasizing the necessity of a quantitative understanding of TF–DNA binding specificities. Structures obtained through X-ray crystallography and NMR spectroscopy allow us to determine the detailed mechanisms of protein–DNA binding involving single DNA target sites and have greatly advanced our perception of protein–DNA recognition (Rohs *et al*, 2010). However, it is inherently difficult to apply these insights at a high-throughput level. Protein crystallization is a time-consuming process, and deriving distance constraints using NMR experiments is costly and likewise time-consuming. As a consequence, structural information is limited to a subset of TFs and individual DNA-binding sites.

In the genomics field, sequencing- and microarray-based high-throughput methods have made it possible to study systematically *in vitro* TF–DNA binding specificities by simultaneously measuring binding affinities to millions of different DNA sequences. *In vitro* platforms such as HT-SELEX and PBM provide effective solutions to gain quantitative knowledge of TF–DNA binding (Berger *et al*, 2006; Zhao *et al*, 2009; Jolma *et al*, 2010), as the confounding factors *in vivo* are not present. With sequencing depth being further improved by an average of 10-fold compared with the original data (Jolma *et al*, 2013), the HT-SELEX data generated in this study currently represent the most extensive set of TF–DNA binding measurements for mammalian TFs. We constructed an analysis pipeline that derives binding affinities for different DNA *M*-words from these HT-SELEX data, gaining a much more detailed view of the binding energy landscape than simple PWM models. This approach enabled us to explore, through statistical machine-learning methods, how the mechanisms of DNA shape readout are employed by various TF families. With feature-selection techniques, we revealed TF family-specific positional DNA shape importance at base pair resolution. The results concur with available experimental structures. Overall, this study provides a means to derive binding mechanisms from sequence data without relying on solved structures.

Despite these methodological advances, we see several limitations in our preprocessing of the data. First, while increasing the sequencing depth improved statistical robustness of binding affinities derived for the short *M*-words used here, the amount of sequencing data may still be insufficient for models using longer *M*-words. Although the sequencing depth could be increased further

(Slattery *et al*, 2011), this endeavor would be expensive, considering the large number of TFs that were studied. Second, HT-SELEX technology can be influenced by oligonucleotide synthesis and PCR bias. In addition, TFs may bind in different binding modes, resulting in enrichment of a mixture of oligonucleotides containing one or more binding motifs. To identify features of single binding events, we based our analysis on known core motifs, allowing only one core motif within each oligonucleotide, removing PCR duplicates, and normalizing by the initial round.

Moreover, we note the limitations in the shape readout profiles and their visualization. First, DNA shape alone is obviously insufficient to explain TF binding (Zhou *et al*, 2015). Second, the shape logos are not equivalent to sequence logos, as they are based on positional scores that do not represent a probability distribution or energy parameters. An alternative way for generating DNA shape logos is to use feature weights derived from models. However, due to the interdependencies between features, such weights are not directly interpretable. In our analysis, we gauged the importance of each individual DNA shape feature by adding it to the 1mer baseline model and observed its effect on the model performance. We believe that DNA shape logos based on such extensive computation are more robust. Although such logos do not yet lead to the prediction of a protein–DNA structure model, they are a step forward and provide a general guide for revealing DNA shape preferences. Third, although the TF–DNA structures supported the heat map results, the correlation is not at all conclusive. Experimentally solved structures in the PDB are not available for most of the studied TFs. Both the "red" and "blue" heat maps aim to summarize the DNA shape importance at individual positions. However, the red heat maps can contain false-positive cells, and the blue heat maps can contain both false-positive and false-negative ones (see definition in Results section). The DNA shape features at a position essentially reflect the pentamer context at that position. Shape features of adjacent positions may contain redundant information. As a result, a position indicated as important in the red heat map may be due to the fact that DNA shape features at the position adjacent to it are important, inducing false positives in the heat map (Appendix Fig S4A and H). On the other hand, for the same reason, a position may not be indicated as important in the blue heat map due to the fact that its directly adjacent position is making up for it, inducing false negatives in the heat map (Appendix Fig S4A and H). Moreover, the DNA shape features used here are derived from sequence, so a position indicated as important in the blue heat map may be due to the loss of sequence information encoded indirectly in the shape features, inducing false positives in the heat map, for example, TBX15 in Appendix Fig S4J. The combined version of the heat maps improves the accuracy to some extent. In addition, the feature-selection analysis that breaks down the DNA shape contribution into individual DNA shape features helps locate the effective shape features. Despite these limitations, we believe that in the future, such heat map analysis, when combined with TF–DNA binding measurements of improved quality, will allow us to gain more clues of TF-binding mechanisms from DNA sequencing data.

Finally, although understanding of *in vitro* protein–DNA binding mechanisms is a critical step toward understanding *in vivo* binding, the *in vivo* scenario consists of multiple layers of complexity, such as the three-dimensional genomic architecture (Rao *et al*, 2015), DNA accessibility (Neph *et al*, 2012), nucleosome competition

(Barozzi *et al*, 2014), and TF cooperativity and co-factors (Slattery *et al*, 2011; Crocker *et al*, 2015). Full understanding of gene regulation will require the integration of knowledge obtained in different fields using various technologies.

In conclusion, while the DNA sequence describes opportunities to form hydrogen bonds and other direct contacts between amino acids and bases, DNA shape can provide an important additional contribution to TF binding (Rohs *et al*, 2009, 2010). We systematically explored here, we believe for the first time, the role of DNA shape readout for many TF families, using high-quality HT-SELEX data, and obtained results at base pair resolution. We produced a valuable TF–DNA binding data resource by increasing the sequencing depth of previous HT-SELEX experiments (Jolma *et al*, 2013) and developing tools for deriving TF–DNA binding affinities and mechanisms from DNA sequencing data.

# Materials and Methods

### HT-SELEX binding data

HT-SELEX experiments were comprised of previously published data (Jolma *et al*, 2013) complemented by new sequencing data. The new data were produced by repooling existing PCR-amplified SELEX ligands into new Illumina sequencing libraries, where samples were multiplexed to a lesser extent (~55× vs. ~800×) than in the previous study. Libraries were sequenced using the Illumina Hiseq2 platform, as in the previous study (Jolma *et al*, 2013). The additional sequencing coverage used in the analysis has been submitted in the European Nucleotide Archive (ENA; http:// www.ebi.ac.uk/ena) under study identifier PRJEB14744. The complete dataset comprises 548 experiments covering 410 different TFs, including mouse/human full-length protein–DNA binding domain differences. Forty protein families were represented. Protein family membership can be found in Jolma *et al* (2013). For the three TFs in the validation set, new HT-SELEX experiments were performed essentially as described in (Nitta *et al*, 2015).

The gcPBM data were downloaded from GEO accession number GSE59845 (Zhou *et al*, 2015). Max protein 12-word scores were the average log-normalized fluorescence intensities of probe sequences that included these 12-words.

### Choosing core motifs

For each TF, we defined a core-binding sequence to enable identification of the most likely binding site and filter out unbound oligonucleotides. We used the seeds published in Jolma *et al* (2013) as the core motifs, but removed their flanks. To pinpoint the core positions as opposed to the flanks, we used motifs compiled in Weirauch and Hughes (2011), which are consensus sequences for only the core motifs collected for different TF families. Substring positions that have the most agreement to any of the corresponding Weirauch and Hughes motifs (Weirauch & Hughes, 2011) were chosen as the core positions. We used the IUPAC character representation for nucleotide sequence. It was sufficient for positions to agree if they represented the same nucleotide. When using the Weirauch and Hughes (Weirauch & Hughes, 2011) motifs as core seeds, most TF families had only one core motif, which would be the assigned motif for TFs

from the family. For TF families having several motifs, we compared the Weirauch and Hughes motifs (Weirauch & Hughes, 2011) to the published consensus seeds (Jolma *et al*, 2013) and calculated *score1*, the portion of matched nucleotides. The core motif with the highest *score1* was assigned to a TF, respectively. If multiple options remained after this step, then we calculated *score2*, a stricter similarity score such that the IUPAC symbols matched exactly (e.g., R matches R but not A). The core motif with the highest *score2* was then selected. This process ensured that almost all TFs were assigned only one motif. In some rare cases, two motifs survived. For both Jolma *et al* seeds and Weirauch and Hughes motifs, when multiple seeds were selected, a dataset for the TF was derived according to each selected seed, but only the dataset with highest $R^2$ was included in the analysis in Fig 3. For a complete list of datasets, see Table EV1.

A few TF families were not covered by Weirauch and Hughes (2011). For C2H2 TFs, we used the seeds published in Jolma *et al* (2013) without removing the flanks, as zinc fingers bind different sequences based on the specificity of each finger (noted in Weirauch and Hughes, 2011). For six TF families not covered by (Weirauch & Hughes, 2011), we used other published resources for the seed of each family, as specified here: RRM (Fernandez-Miranda & Mendez, 2012), NFI (Whittle *et al*, 2009), NRF (http://AtlasGeneticsOncol ogy.org), TFAP (http://AtlasGeneticsOncology.org), and znf_BED (http://www.genecards.org). For the complete list of core consensus motifs, see Table EV3.

### M-word scores

We derived *M*-word binding scores based on observed experimental enrichment counts. HT-SELEX experiments included several rounds of enrichment of bound DNA sequences by a specific protein. Initially, the experiment began from a pseudo-random DNA oligonucleotide library. The protein was allowed to bind to DNA sequences in the randomized pool. Next, bound ("selected") sequences were isolated and amplified for sequencing and reiteration of the process. The frequency of DNA sequences that have higher binding affinities increased exponentially. It is possible to derive the binding affinity for DNA sequences based on their change in frequencies throughout the rounds (Levine & Nilsen-Hamilton, 2007). In the HT-SELEX experiments, the oligonucleotide length (excluding constant ends) was 14, 20, 30, or 40 base pairs.

*M*-word scores were produced for each core motif, with the following parameters: number of core-flanking positions to derive, selected round, and number of core mismatches that were allowed. For each HT-SELEX oligonucleotide, at most one BS was accounted for. An *M*-word with a number of matches to the core motif above the threshold was chosen as the BS (if there were several, the oligonucleotide was discarded to avoid multiple modes of binding). Only for occurrences in which the *M*-word had sufficiently long flanks to include, the required side positions were used. The reverse complement strand was also considered and, in cases of hits on both strands, the one with the larger number of matches was used. If no *M*-word matched the core motif given the allowed number of mismatches, the oligonucleotide was discarded.

To produce accurate *M*-word ratio scores, counts were divided by estimated frequencies in the initial pool, as previously described (Slattery *et al*, 2011). Estimated frequencies were generated using a fifth-order Markov model of observed frequencies in the initial pool,

following the SELEX-seq protocol (Slattery *et al*, 2011). The score was the *i*th root of the ratio, where *i* was the round of selection. This approach was based on the assumption that *M*-word frequencies increased by the same factor between two consecutive selection cycles (Slattery *et al*, 2011). To compare different alternative scores, we considered the frequency at round *i* and the ratio of the frequency at round *i* over the (observed) frequency in the initial round. In all cases, an oligonucleotide was only counted once to avoid PCR duplication bias.

### Length of core-motif and flanking regions, number of mismatches allowed, and selected rounds

For each experiment, *M*-word scores were derived per round for round 3 and later rounds. As the first few rounds did not show a profound enrichment, we did not consider them. Later rounds showed enrichment and varied in quality and read depth. Thus, data were collected per round from round 3 onwards, and selection of the round was deferred to a later stage.

Similarly, we generated datasets for different values of *M*. There is an inherent tradeoff between increasing *M* and reducing the accuracy of the scores. While greater *M* values provide information on binding to longer flanks, counts of *M*-words decrease as *M* grows, leading to less accurate binding scores. Keeping this tradeoff in mind, we considered the initial length and the maximum length of flanking regions. The initial length was set to $\lfloor (10 - core\_length)/2 \rfloor$ so that *M* is at least 10, allowing DNA shape prediction for at least 6 positions (the two positions at each flank are not available due to the pentamer model). For example, for core TAAATTA of length 7, the initial flank length was 1. We called an *M*-word reliable if its count was > 8. *M* was set to be the largest value for which the number of reliable *M*-words was ≥ 1,000, and the maximum *M*-word count was ≥ 100. When all *M*-word counts are < 100, the scores may be inaccurate, and samples with less than 1,000 reliable *M*-words are considered small and excluded from our analysis. For example, for the same core of TAAATTA, if GAGTAAAT-TACTC was the most frequent 13-word and it appeared only 89 times, whereas the 11-word AGTAAATTACT appeared 1,540 times, assuming there are more than 1,000 reliable *M*-words in both, the maximum length would be 3 (the core is of length 7, leaving 3 flanking positions on each side). Datasets were created for all flanking region lengths, starting at the initial and up to the maximum length.

Another tradeoff exists in the number of mismatches: up to a point, allowed mismatches increase the variability of *M*-words, and thus add useful information. Too many mismatches would lead to the introduction of *M*-words that do not represent BSs, resulting in added noise. With this tradeoff in mind, we set the number of mismatches allowed to depend on the length of the core motif. Generally, the number was $\lfloor (core\_length - 4)/2 \rfloor + 1$. In case the core motif contains degenerate characters, that is, those that represent multiple nucleotides, we counted these characters differently in the core length. The weight of a character in this count was 1/*nucleotides_it_represents*, and the length of a core was the sum of its characters' weights. For example, for ATAAAA, we allowed two mismatches as there are six characters of weight 1. For CANNTG, we allowed only one mismatch (in addition to the two central fully degenerate positions), because its total weight is 4*1+2*0.25 = 4.5. By applying this threshold, on average, 74 ± 25% of the oligonucleotides were retained, which suggests that it can detect probable BSs while

removing oligonucleotides that are less likely to be bound. The above threshold was used as a first step in order to exclude unbound oligonucleotides. In the second step, a stricter threshold allowing one less mismatch was used to filter out oligonucleotides that have multiple motif occurrences, in order to exclude cooperative binding events from our analysis. The stricter threshold ensures that not too many oligonucleotides are filtered out in the second step. Finally, the oligonucleotides were aligned according to the core motif.

### Dataset filtering

In large-scale experimental data, it is inherently difficult to ensure that every dataset has equivalent diversity and enrichment level. Although PWM models can be constructed from low-quality experimental data, complex models require high levels of enrichment and sequence diversity. To reach reliable conclusions, we used multiple data filtering procedures to discard datasets of insufficient quality. We performed two stages of QC for these datasets. In the first stage, we used four QC criteria to ensure high counts for accurate score estimates, large sample size, and score variability.

1. All *M*-words with count ≤ 8 were discarded because low counts lead to inaccurate estimates of binding scores.
2. If the number of different *M*-words after step 1 was < 1,000, then the dataset was filtered out, to ensure that datasets have sufficient numbers of samples for the learning algorithm.
3. Datasets were tested for variable scores. The score of the $90^{th}$ percentile had to be at least 0.2 greater than the score of the $10^{th}$ percentile.
4. The maximum *M*-word had to appear at least 100 times; otherwise, counts would be too small and estimates inaccurate for most of the *M*-words.

We filtered out datasets based on $R^2$ performance criteria. We ran L2-regularized multiple linear regression (MLR) on each of the remaining datasets using different combinations of features. Due to their linearity, we would expect that, for MLR models, model A would perform at least as well as model B, given that B uses a subset of features used by A. We defined a dataset as invalid only when the performance of model A was smaller than that of model B by more than 3%, given that B uses a subset of features used by A. This process reduced the number of valid datasets to 533. Datasets for which even the best model had $R^2 < 0.5$ were excluded from the analyses. Finally, 512 datasets covering 215 human/mouse TFs belonging to 27 different TF families passed our QC procedure.

For TFs covered by multiple datasets, only the dataset with the highest $R^2$ was included in downstream analyses (see Table EV1 for the complete list). As PCA requires only one representative BS sequence for each TF, we separately generated 12-word data using reads from the last round of HT-SELEX, as the last round is expected to be the most specific. We used the top 12-word as the representative BS for each TF. In doing so, as many as 294 TFs were covered in the PCA (Fig 2; see Table EV4 for a complete list of 12-words for the 294 TFs in the PCA).

### PCA and linear regression analysis

For each DNA sequence *s*, the 1-mer, 2-mer, and 3-mer features were encoded into feature vectors $\phi^{1mer}$, $\phi^{2mer}$, and $\phi^{3mer}$, respectively, in a similar way to those used in Zhou *et al* (2015). The $i^{th}$

nucleotide in $s$ was denoted $s^i$. Elements of vectors $\phi^{1mer}$, $\phi^{2mer}$, and $\phi^{3mer}$ were formulated as follows. For nucleotide position $i$:

$$\phi^{1mer}_{4*(i-1)+1}(s) = \begin{cases} 0, & \text{if } s^i \neq A \\ 1, & \text{if } s^i = A \end{cases}, i = 1, \ldots, l$$

$$\phi^{1mer}_{4*(i-1)+2}(s) = \begin{cases} 0, & \text{if } s^i \neq C \\ 1, & \text{if } s^i = C \end{cases}, i = 1, \ldots, l$$

$$\phi^{1mer}_{4*(i-1)+3}(s) = \begin{cases} 0, & \text{if } s^i \neq G \\ 1, & \text{if } s^i = G \end{cases}, i = 1, \ldots, l$$

$$\phi^{1mer}_{4*i}(s) = \begin{cases} 0, & \text{if } s^i \neq T \\ 1, & \text{if } s^i = T \end{cases}, i = 1, \ldots, l$$

$$\phi^{2mer}_{16*(i-1)+1}(s) = \begin{cases} 0, & \text{if } s^i s^{i+1} \neq AA \\ 1, & \text{if } s^i s^{i+1} = AA \end{cases}, i = 1, \ldots, l-1$$

$$\phi^{2mer}_{16*(i-1)+2}(s) = \begin{cases} 0, & \text{if } s^i s^{i+1} \neq AC \\ 1, & \text{if } s^i s^{i+1} = AC \end{cases}, i = 1, \ldots, l-1$$

$$\phi^{2mer}_{16*(i-1)+3}(s) = \begin{cases} 0, & \text{if } s^i s^{i+1} \neq AG \\ 1, & \text{if } s^i s^{i+1} = AG \end{cases}, i = 1, \ldots, l-1$$

$\ldots$

$$\phi^{2mer}_{16*i}(s) = \begin{cases} 0, & \text{if } s^i s^{i+1} \neq TT \\ 1, & \text{if } s^i s^{i+1} = TT \end{cases}, i = 1, \ldots, l-1$$

$$\phi^{3mer}_{64*(i-1)+1}(s) = \begin{cases} 0, & \text{if } s^i s^{i+1} s^{i+2} \neq AAA \\ 1, & \text{if } s^i s^{i+1} s^{i+2} = AAA \end{cases}, i = 1, \ldots, l-2$$

$$\phi^{3mer}_{64*(i-1)+2}(s) = \begin{cases} 0, & \text{if } s^i s^{i+1} s^{i+2} \neq AAC \\ 1, & \text{if } s^i s^{i+1} s^{i+2} = AAC \end{cases}, i = 1, \ldots, l-2$$

$$\phi^{3mer}_{64*(i-1)+3}(s) = \begin{cases} 0, & \text{if } s^i s^{i+1} s^{i+2} \neq AAG \\ 1, & \text{if } s^i s^{i+1} s^{i+2} = AAG \end{cases}, i = 1, \ldots, l-2$$

$\ldots$

$$\phi^{3mer}_{64*i}(s) = \begin{cases} 0, & \text{if } s^i s^{i+1} s^{i+2} \neq TTT \\ 1, & \text{if } s^i s^{i+1} s^{i+2} = TTT \end{cases}, i = 1, \ldots, l-2$$

First-order DNA shape features MGW, ProT, Roll, and HelT, denoted $\phi^{MGW}$, $\phi^{ProT}$, $\phi^{Roll}$, and $\phi^{HelT}$, respectively, were generated by our DNAshape prediction method (Zhou *et al*, 2013; Chiu *et al*, 2016). For these DNA shape features, the following normalization was performed:

$$\phi^{MGW}_i = (MGW_i - MGW_{min})/MGW_{sd}$$

where $MGW_i$ is the predicted MGW, $MGW_{min}$ is the minimum MGW over all possible pentamers, and $MGW_{sd}$ is the standard deviation of MGW in the data. Similarly:

$$\phi^{ProT}_i = (ProT_i - ProT_{min})/ProT_{sd},$$
$$\phi^{Roll}_i = (Roll_i - Roll_{min})/Roll_{sd},$$
$$\phi^{HelT}_i = (HelT_i - HelT_{min})/HelT_{sd}.$$

Second-order DNA shape features were derived from the first-order features and denoted $\phi^{MGW^2}$, $\phi^{ProT^2}$, $\phi^{Roll^2}$, and $\phi^{HelT^2}$. These second-order shape features were the product terms of adjacent first-order DNA shape features, normalized by the standard deviation. MGW and ProT were defined for each base pair, and Roll and HelT were defined for each base pair step. Thus, in the feature-selection analysis, DNA shape features at nucleotide position $i$, denoted as $shape_i$, consisted of $\phi^{MGW}_i$, $\phi^{ProT}_i$, $\phi^{Roll}_i$, $\phi^{Roll}_{i+1}$, $\phi^{HelT}_i$, $\phi^{HelT}_{i+1}$, $\phi^{MGW^2}_i$, $\phi^{MGW^2}_{i+1}$, $\phi^{ProT^2}_i$, $\phi^{ProT^2}_{i+1}$, $\phi^{Roll^2}_i$, and $\phi^{HelT^2}_i$. If the core-motif sequence

was palindromic, then the last step in the feature encoding was to symmetrize the feature vector by averaging it with the feature vector encoding the reverse complementary stand. The DNAshape method predicts shape features based on a pentamer query table that is derived from all-atom Monte Carlo simulations (Zhou *et al*, 2013). As a control, we shuffled the pentamer query table and tested its effects on shape models.

After the feature encoding, L2-regularized MLR and 10-fold cross-validation were performed for each dataset to gauge model performance (Yang *et al*, 2014; Abe *et al*, 2015). L2-regularized MLR was chosen for its simplicity and interpretability. In PCA, the feature vector encoded for the sequence of highest DNA-binding affinity of a TF was used to represent that TF.

### 3mer and 1mer+2mer+3mer model equivalence in linear regression

The 3mer models and 1mer+2mer+3mer models are equivalently "powerful" in MLR, where the power of a model refers to its descriptive capability. This equivalency can be demonstrated by showing that any solution of a 1mer+2mer+3mer model could be mapped into a 3mer model solution that gives exactly the same prediction of binding affinity for any input DNA sequence, and *vice versa*. Proof for the reverse direction is trivial. We could just keep the learned coefficients, or weights, of the 3-mer features, and set all weights for 1-mer and 2-mer features to be zero. This process results in a 1mer+2mer+3mer model that gives exactly the same prediction of binding affinity for any input DNA sequence as the original 3mer model. Mapping for the other direction is as follows.

Denote a solution to a 1mer+2mer+3mer model as:

$$S_1 = (w^1_A, w^1_C, w^1_G, w^1_T, w^2_A, w^2_C, w^2_G, w^2_T, \ldots,$$
$$w^{N-1}_A, w^{N-1}_C, w^{N-1}_G, w^{N-1}_T, w^N_A, w^N_C, w^N_G, w^N_T,$$
$$w^1_{AA}, w^1_{AC}, w^1_{AG}, w^1_{AT}, w^1_{CA}, w^1_{CC}, w^1_{CG}, w^1_{CT}, \ldots,$$
$$w^{N-1}_{GA}, w^{N-1}_{GC}, w^{N-1}_{GG}, w^{N-1}_{GT}, w^{N-1}_{TA}, w^{N-1}_{TC}, w^{N-1}_{TG}, w^{N-1}_{TT},$$
$$w^1_{AAA}, w^1_{AAC}, w^1_{AAG}, w^1_{AAT}, w^1_{ACA}, w^1_{ACC}, w^1_{ACG}, w^1_{ACT}, \ldots,$$
$$w^{N-2}_{TGA}, w^{N-2}_{TGC}, w^{N-2}_{TGG}, w^{N-2}_{TGT}, w^{N-2}_{TTA}, w^{N-2}_{TTC}, w^{N-2}_{TTG}, w^{N-2}_{TTT}).$$

Denote a solution to the 3mer model as:

$$S_2 = (m^1_{AAA}, m^1_{AAC}, m^1_{AAG}, m^1_{AAT}, m^1_{ACA}, m^1_{ACC}, m^1_{ACG}, m^1_{ACT}, \ldots,$$
$$m^{N-2}_{TGA}, m^{N-2}_{TGC}, m^{N-2}_{TGG}, m^{N-2}_{TGT}, m^{N-2}_{TTA}, m^{N-2}_{TTC}, m^{N-2}_{TTG}, m^{N-2}_{TTT}).$$

Superscript numbers denote nucleotide positions in the DNA sequences. Subscript letters denote what features at those positions the learned weights are for. For any $x, y, z \in \{A, C, G, T\}$, map the weights as follows:

$$m^i_{xyz} = w^i_{xyz} + w^i_{xy} + w^i_x, i = 1, \ldots, N-3$$
$$m^{N-2}_{xyz} = w^{N-2}_{xyz} + w^{N-2}_{xy} + w^{N-2}_x + w^{N-1}_{yz} + w^{N-1}_y + w^N_z.$$

The resulting $S_2$ will assign the same predicted binding affinity as $S_1$ to any input DNA sequence.

The equivalency between 3mer and 1mer+2mer+3mer models no longer holds strictly when regularization is added. It is only true if we assume that the training process always ensures that the learned

model has the highest generalization accuracy under the MLR framework, that is, the optimal solution. In practice, the solution is not necessarily the optimal one, despite being the goal of the regularization. Thus, 1mer+2mer+3mer models and 3mer models are approximately equivalent in the L2-regularized MLR used here. For this reason, we see that the data points drifted slightly off the diagonal in Appendix Fig S3F.

**Generating DNA shape logos**

DNA shape logos were generated using the seq2logo program with the PSSM-logo option (Thomsen & Nielsen, 2012). We gauged the importance of each DNA shape feature at each nucleotide position by adding this feature to the baseline 1mer model. We then calculated the $\Delta R^2$ value upon adding this particular feature. These $\Delta R^2$ values were used to construct a position-specific scoring matrix (PSSM), which served as input to the seq2logo program. DNA sequence logos were generated based on PSSMs that were calculated from top 200 *M*-words for each TF.

**Data availability**

The raw sequencing data from the HT-SELEX experiments are available at ENA (http://www.ebi.ac.uk/ena) under study identifier PRJEB14744. All MLR models and PSSMs are available at BioStudies (https://www.ebi.ac.uk/biostudies) under accession number S-BSST6 or at http://rohslab.cmb.usc.edu/MSB2017.

**Expanded View** for this article is available online.

## Acknowledgements

This work was performed in part while Y.O. and R.S. were visiting the Simons Institute for the Theory of Computing at UC Berkeley. The work was supported by the National Institutes of Health (grants R01GM106056 and U01GM103804 to R.R.) and an Alfred P. Sloan Research Fellowship (to R.R.), the Israel Science Foundation (grant 317/13 to R.S.) and the Raymond and Beverly Sackler Chair in Bioinformatics (to R.S.), and Knut and Alice Wallenberg Foundation and Swedish Research Council grants (to J.T.). L.Y. and Y.O. acknowledge support through Dan David Prize scholarships. Y.O. was partially supported by the Edmond J. Safra Center for Bioinformatics at Tel Aviv University. Open-access charges were defrayed in part through the National Science Foundation (grant MCB-1413539 to R.R.).

## Author contributions

LY, YO, RS, and RR conceived and designed the project and wrote the paper. LY and YO developed computational methods and analyzed data. AJ, YY, and JT generated HT-SELEX data. JT directed the HT-SELEX experiments. RS and RR directed the computational study and overall project.

## Conflict of interest

The authors declare that they have no conflict of interest.

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
