## [Review Process File · Molecular Systems Biology]

Transcription factor family-specific DNA shape readout revealed by quantitative specificity models

Lin Yang, Yaron Orenstein, Arttu Jolma, Yimeng Yin, Jussi Taipale, Ron Shamir and Remo Rohs

Corresponding author: Remo Rohs, University of Southern California

Review timeline:

Submission date:	04 August 2016
Editorial Decision:	26 August 2016
Revision received:	24 November 2016
Editorial Decision:	20 December 2016
Revision received:	04 January 2017
Accepted:	05 January 2017

Editor: Maria Polychronidou

Transaction Report:

1st Editorial Decision

26 August 2016

Thank you again for submitting your work to Molecular Systems Biology. We have now heard back from the three referees who agreed to evaluate your study. As you will see below, the reviewers raise a number of concerns, which unfortunately preclude the publication of the study in its current form.

The reviewers mention that i) as it stands the main conclusions are not well supported and that ii) the novel aspects of the work, that would represent an advance compared to previous related studies, are not sufficiently expanded. However, considering that the reviewers appreciate that the study presents a rather comprehensive analysis of the effect of DNA shape across TFs and is therefore likely to be useful for the field, we would like to offer you a chance to revise the study and address the points raised.

Without repeating all the points/comments listed below, the most fundamental issues that need to be convincingly addressed are the following:

- Further analyses are required to better support the main conclusions. Reviewer #1 provides constructive suggestions related to this point.

- As reviewer #2 mentions, the main novel aspect of the work is analyzing the role of DNA shape across TF families. Therefore, the study will benefit from extending the related analyses (and discussion) beyond homeodomain TFs. Moreover, expanding the last section of the manuscript, describing how DNA shape features can provide mechanistic insights, would significantly enhance the impact of the study.

Of course, all other issues listed by the referees need to be addressed.

REFeree REPORTS

Reviewer #1:

This manuscript explores the contribution of DNA shape to transcription factor-DNA binding models. The authors train machine-learning models of TF-DNA binding specificities using data from HT-SELEX of 197 TFs. By comparing models with different features the authors make the following claims: 1) DNA shape contributes to TF-DNA binding preferences across many TF families, 2) DNA shape features in core-motif flanks are important for many TF families, 3) DNA shape contribution has TF-family-specific positional preferences within binding sites and 4) DNA shape features provide mechanistic insights into TF-DNA binding.

Overall, the data only weakly support the claims presented in this manuscript.

With respect to the first claim, Figure 3A shows that adding shape features to a 1mer model improves R2 for at least half of the TFs studied. However, it is not clear if the improvement is because of shape features or because of the increased number of parameters. The authors could distinguish between the two by randomizing pentamer identity-value associations of shape features (as done in Zhou et al, 2015) and provide stronger support for claim 1. Another concern is that in Supplemental figure S3, it seems that the Weirauch & Hughes motif seeds have higher DNA-shape induced performance gain compared to Jolma et al. 2013 seeds. Using the Jolma et al seeds could decrease the number of TFs that show significant shape dependence. This is an issue since there is no real justification for using the Weirauch & Hughes seeds over the Jolma seeds.

With respect to the first and second claims the authors infer from Figure 3B that 1mer + shape models do not perform as well as 1mer+2mer+3mer models because the edge features cannot be incorporated into shape features. They test this hypothesis by adding edge features (3merE2) to the 1mer+shape models. From figures 3C and 3D it is not evident whether 3merE2 features represent missing shape features or simply an increased number of parameters. Another test would be to remove the edge features from the 1mer+2mer+3mer model and compare it to the 1mer+shape model.

The strongest evidence for the importance of shape features in core-motif flanks (claim 2) is in Supplemental Figure S4, however it is limited by small sample size.

With respect to the third claim, Figure 5C and Supplemental Figure S6 show TF-specific positional preferences of DNA shape features. However, it is hard to discern any family-specific patterns from the heat maps. It is also hard to make an assessment of claim 3, because important information about the contributions of individual shape features (Roll, Propeller Twist, Helical Twist and Minor Groove Width) is lost in this representation of the data. Perhaps some form of clustering of heat maps would help the readers appreciate TF-family specific positional preferences of DNA shape.

A major claim of this manuscript is that DNA shape features can provide mechanistic insight into TF-DNA binding. This claim is largely unsupported. Firstly, the heat maps in Figures 5C and S6 depict the contribution of shape features to a machine learning model. The relationship between a feature weight and contribution to TF-DNA binding preferences is indirect. Secondly, the features are not broken down into individual DNA shape features, making it harder to predict mechanism from shape contribution.

The concept of "shape logos" sounds interesting but there are issues with this representation. Sequence logos depict contributions of each base at every position of a TFBS. A shape logo cannot achieve this because the height of the letters represents features weights from a machine-learning model, not contributions to TF binding explicitly. Also, different shape features are not independent of each other in the same way the nucleotides at a certain position are independent. In the PITX3 example shown by the authors, it would be impossible to predict from the logo (or the heat map) that the PITX3-DNA structure contains a narrow minor groove region on the 3' side of the core motif.

This work provides DNA-shape based binding models for more transcription factors from diverse

families compared to previous work (Zhao et al, 2015). The most interesting part of the manuscript for me was that shape models achieve similar performance as 3mer based models with far fewer parameters. Overall, the first three claims are currently weakly supported. The last claim about mechanistic insight is not supported and should be moved to the discussion section.

Minor comment:

1. Legend for Figure S4 is incorrect.

Reviewer #2:

Yang, et al. present a comprehensive analysis of DNA shape readout across a large set of mammalian transcription factors spanning several structural families. The work is based on resequencing (and thereby extending) 548 previously published HT-SELEX experiments, which provide a quantitative metric of in vitro TF-DNA binding affinity across a large number of DNA sequences. These data were used to train regression models that use positional sequence information alone or sequence plus DNA shape parameters to predict binding affinity. The importance of DNA shape readout across various TFs (and at various positions within the TF binding site) is mainly analyzed via the relative performance of regression models that are trained with different combinations of sequence and shape data. The work concludes that shape readout is important for DNA binding recognition across many TF families, although the relative positions affected by shape readout vary within and between TF structural families.

General comments:

Methodologically, this work does not substantially extend on the authors' previous publications. While the HT-SELEX resequencing has produced a more statistically robust data resource that will be of use to the broader community, this effort does not qualitatively change the nature of the information gained on the profiled TFs. Similarly, the computational methods in this work are straightforward applications of methods previously presented in Zhou, et al. 2015 and Abe, et al. 2015. That said, it is worth noting that the analyses presented in Fig 3 nicely reconcile a point of confusion in the Zhou, et al. manuscript, and now more firmly confirm that DNA shape parameters provide an efficient encoding of higher-order sequence information at binding sites. The greater significance of this work centers on its survey of DNA shape influences across a large number of TFs from various structural classes. As far as I am aware, this is the most comprehensive examination of the relative importance of DNA shape readout across diverse TFs to date.

Major points

1) Given that the main novelty of this work is in the survey of DNA shape readout across TF families, it is disappointing that more interpretation and discussion is not focused on these results. For example, it may have been informative to move beyond homeodomains to discuss in more detail other TF families and how DNA shape influences their binding affinity (perhaps referencing representative protein-DNA structures for each family).

2) One worrying aspect of the presented survey in Figs 5 & S6 was that the two feature selection strategies (adding shape info versus removing shape info at position i in the regression model) provide several conflicting results. For example, the two strategies point to shape being important on opposite sides of the core motif for Lhx4 and others in Fig 5, and yield opposing conclusions on the positional influence of DNA shape for several families in Fig S6 (e.g. bHLH, bZIP, POU). How can these conflicts be explained/reconciled?

3) I'm confused about how Fig 6 is claimed to support the conclusions derived from Fig 5. It may be helpful if the bases are labeled in Fig 6 A & B, but it seems that the narrowing MGW occurs at the center of the core motif, and not at the 3' side of the motif as expected from the heatmap in Fig 5. Similarly, and for the same reasons, I would have expected to see larger feature weights at positions 7 & 8 in the DNA shape logos presented in Fig 6 C & D.

4) It's unfortunate that only one protein (Max) has been characterized by both gcPBM and HT-SELEX, as it leaves only one dataset with which to assess the marginal value of deeper HT-SELEX

coverage. Are there any alternate metrics with which to assess this point?

5) I'm not sure how to interpret the PCA-based analyses. While the first two principal components in Fig 2B (1-mers + shape) account for more variance than those in Fig 2A (1-mers alone), are the two PCA projections truly comparable in this way? Is it possible that the greater amount of explained variance is due somehow to the differences in dataset dimensionality? The intra- & inter-family distance analyses based on the PCA plots are similarly difficult to interpret. It's claimed that the difference in median inter and intra-family group distances is larger when DNA shape features are incorporated in the model. However, that only seems true in absolute (as opposed to relative) terms.

Minor points

6) p.7 states that the regression model used is similar to the methodology in Zhou, et al. 2015. However, L2-regularized MLR is used in this work, whereas an SVR approach was used in Zhou, et al. While these methods perform similarly, it may be more correct to point to the Yang, et al. 2014 or Abe, et al. 2015 instead.

7) I find the new DNA shape logos quite informative. One problem with this representation is the lack of natural limits on the y-axes. Can any guidance be provided on that aspect?

Reviewer #3:

Yang, et al. describe a large-scale computational analysis to uncover the contribution of DNA shape to transcription factor (TF) binding. To carry out this analysis, they use HT-SELEX data describing the binding preferences of hundreds of mammalian TFs and augment this data set with increased sequencing depth. They use models that describe TF binding preferences with and without shape parameters to show that shape is important for many families of TFs and use two approaches to identify specific positions at which shape is important. Finally, they introduce a sequence logo to display TF shape preferences.

Over the last few years, the importance of DNA shape in TF binding has been highlighted by several papers, including many by Rohs, one of the corresponding authors of this paper. The novelty of this paper comes from two features: (1) the authors perform an analysis of the role of DNA shape on a larger set of TFs, representing more DNA binding domain families, than previous studies (e.g. Zhou, et al. 2015 PNAS), and (2) they measure base pair-specific information about the contribution of DNA shape to TF binding and encode this information in a DNA shape logo. This work should be of interest to large community of scientists interested in understanding TF-DNA interactions. The conclusions in the paper are generally well-supported by the data, but some parts of the text are difficult to understand, making some claims harder to evaluate. The figures are generally clear and easy to interpret. The authors did a nice job of providing a comprehensive amount of supplementary data, including the details of all the data sets used, the models for each TF, etc.

Major suggestions:

- For clarity, it would be useful to start paragraph 2 with the goal of selecting the core motif. I got the idea after reading the whole paragraph, but it took a little while. In general, the paper could be vastly improved by a thorough editing of the text. In particular, starting paragraphs with topic sentences that explain either the rationale or goal for the next step of analysis or the connection to the previous step would make the paper far more readable.

- What is the rationale for the formula for choosing the number of allowed mismatches? Do the results depend on the details of this formula?

- What is the rationale for the formula for choosing the initial and maximum length of the flanking regions? Again, do the results depends strongly on the details of this formula?

- It's not clear to me how "good" of a correlation 0.64 is and how the error in the M-word scores will influence the conclusions. It is possible to validate M-word scores for any other proteins using regular PBMs?

- In looking at the shape importance figures in S6, I noticed that between the two feature selection protocols, some blue and red heat maps seemed correlated (e.g. the RFX proteins) and some seemed anti-correlated (e.g. TFAP). What is the significance of this observation? Or in Figure 5, some proteins, e.g. Lhx8 show a correlation between the blue and red heat maps and others, e.g. Lhx4, show an anti-correlation. It would be nice to see a deeper discussion of the significance of the red and blue heat maps when compared to each other.

Minor suggestions:

- The first 3 sentences of the introduction are rather convoluted and could be simplified for clarity.

- It would be helpful to include a sentence in the introduction about the TFs included in the HT-SELEX database. Which mammals are they from? How were they selected?

- In paragraph 1 of results, a total of 197 TFs were said to pass quality control, but in Figure 1, 218 seem to pass the first round of QC.

- page 5, line 7 "Increased sequencing depth will allow us" - most of the paper is written in the past tense. I would keep it consistent here.

- The description of the algorithm to determine maximum length of flanking regions was really hard to understand. (E.g. in the example given, I'm not exactly sure how "3" is the answer.)

- I would highlight more prominently the result in Figure S1C - you tried alternative M-word scoring, which didn't perform as well.

- In the results (page 7), and/or in a figure, it would be useful to enumerate the DNA shape features that are encoded into the feature vectors.

- I think the Methods section could be re-ordered to match the order that each method is used in the results section more closely.

- It would be helpful to have a table with each row corresponding to a TF family, and columns for the number of representatives, fraction that are improved by including shape, average difference between r^2 with and without shape. (This might go in the main text.)

- I didn't understand the claim on page 9 about homeodomains binding as monomers. Was the goal to justify why you would see shape preferences for homeodomains, even as monomers? Or something else? Please clarify.

- Can you include the same details in the text for how you calculated the % change in R^2 for Figure 5B as you did for Figure 5A?

- In general, the figure legends could be expanded to help the reader understand what to take away from each figure. Many of the supplementary figures didn't have any legends, and for some, I really needed them (e.g. Figure S2).

1st Revision - authors' response

24 November 2016

Reviewer #1:

This manuscript explores the contribution of DNA shape to transcription factor-DNA binding models. The authors train machine-learning models of TF-DNA binding specificities using data from HT-SELEX of 197 TFs. By comparing models with different features the authors make the following claims: 1) DNA shape contributes to TF-DNA binding preferences across many TF families, 2) DNA shape features in core-motif flanks are important for many TF families, 3) DNA shape contribution has TF-family-specific positional preferences within binding sites and 4) DNA shape features provide mechanistic insights into TF-DNA binding.

Overall, the data only weakly support the claims presented in this manuscript.

We thank this Reviewer for their frank opinion and think that the revised version of our manuscript addresses the valid concerns raised by this and the other reviewers.

With respect to the first claim, Figure 3A shows that adding shape features to a 1mer model improves R2 for at least half of the TFs studied. However, it is not clear if the improvement is because of shape features or because of the increased number of parameters. The authors could distinguish between the two by randomizing pentamer identity-value associations of shape features (as done in Zhou et al, 2015) and provide stronger support for claim 1. Another concern is that in Supplemental figure S3, it seems that the Weirauch & Hughes motif seeds have higher DNA-shape induced performance gain compared to Jolma et al.2013 seeds. Using the Jolma et al seeds could decrease the number of TFs that show significant shape dependence. This is an issue since there is no real justification for using the Weirach & Hughes seeds over the Jolma seeds.

We agree with the Reviewer and now include an analysis that uses randomly shuffled shape features, compared to the original ones (p. 9, l. 13-16, Fig. S3B). We added a scatter plot comparing a shuffled pentamer query table with the MC-derived query table (as done in Zhou et al, 2015). We also added a PCA analysis using randomly shuffled shape features (p. 8, l. 4-8, Fig. S2). In both cases, we see that the improvement achieved by shape features was diminished when they were shuffled or randomized.

For the second concern, we now base all of our analyses on Jolma et al. seeds (p. 5, l. 19-21, p. 20 l. 17-22). This did not change our previous conclusions that were based on the Weirauch & Hughes motif seeds (p. 9, l. 20-23). We think that showing the robustness with respect to the seed selection was an important addition to the manuscript.

With respect to the first and second claims the authors infer from Figure 3B that 1mer + shape models do not perform as well as 1mer+2mer +3mer models because the edge features cannot be incorporated into shape features. They test this hypothesis by adding edge features (3merE2) to the 1mer +shape models. From figures 3C and 3D it is not evident whether 3merE2 features represent missing shape features or simply an increased number of parameters. Another test would be to remove the edge features from the 1mer+2mer+3mer model and compare it to the 1mer+shape model.

We added the suggested test to strengthen our argument (p. 11 l. 6-8, Fig. S3F). Indeed, we now observe the same phenomenon of comparable performance when omitting shape features at the extreme ends. We think that this test indeed settles the open question in comparing the performance of k-mer and shape features.

The strongest evidence for the importance of shape features in core-motif flanks (claim 2) is in Supplemental Figure S4, however it is limited by small sample size.

We agree with the Reviewer's comment. Unfortunately there are only few gcPBM datasets available to date to test on. All currently available gcPBM datasets for mammalian TFs were included in this study.

With respect to the third claim, Figure 5C and Supplemental Figure S6 show TF-specific positional preferences of DNA shape features. However, it is hard to discern any family-specific patterns from the heat maps. It is also hard to make an assessment of claim 3, because important information about the contributions of individual shape features (Roll, Propeller Twist, Helical Twist and Minor Groove Width) is lost in this representation of the data. Perhaps some form of clustering of heat maps would help the readers appreciate TF-family specific positional preferences of DNA shape.

We share this concern raised by the Reviewer and now attempted to rectify the interpretability by adding an additional heat map based on the aggregation of the two heat maps (p. 13, l. 1-4, Figs 5C, S4). This "overlay" purple heat map combines the original red and blue heat maps in order to reduce noise, i.e. false positives and false negatives of important shape features. The "overlay" heat map is the minimum of the red and blue heat maps for each cell.

To gauge the contribution of individual shape features, we also expanded the feature selection analysis to include testing of the individual contribution of each DNA shape feature category (i.e. MGW, HelT, ProT, and Roll, respectively) separately at each nucleotide position (p. 13, l. 7-9, Fig S5). The resulting heat maps provide a more detailed guide on the importance of each DNA shape feature at every position. To the best of our knowledge, such a comprehensive analysis has never been done in the field.

A major claim of this manuscript is that DNA shape features can provide mechanistic insight into TF-DNA binding. This claim is largely unsupported. Firstly, the heat maps in Figures 5C and S6 depict the contribution of shape features to a machine learning model. The relationship between a feature weight and contribution to TF-DNA binding preferences is indirect. Secondly, the features are not broken down into individual DNA shape features, making it harder to predict mechanism from shape contribution.

We fully understand the Reviewer's criticism with respect to this claim and in response modified the shape logo definition (p. 15, l. 11-19, Figs 6C, D). We no longer use feature weights for generating DNA shape logos, because the Reviewer makes a correct point that the relationship between a feature weight and contribution to binding is indirect. Instead, we now use the relative R^2 gain in percentage as the height for the logos, in line with the heat map visualization.

We also followed the Reviewer's suggestion regarding the need for analyzing contributions of individual shape feature categories (p. 13, l. 7-9, Fig S5). The shape effects are now broken into individual DNA shape feature categories, which provide us not only with the relative contribution of different types of DNA shape features (i.e. MGW vs. Roll vs. ProT vs. HelT) but also with a detailed map of how much and where DNA shape features are affecting TF binding specificity.

The concept of "shape logos" sounds interesting but there are issues with this representation. Sequence logos depict contributions of each base at every position of a TFBS. A shape logo cannot achieve this because the height of the letters represents features weights from a machine-learning model, not contributions to TF binding explicitly. Also, different shape features are not independent of each other in the same way the nucleotides at a certain position are independent. In the PITX3 example shown by the authors, it would be impossible to predict from the logo (or the heat map) that the PITX3-DNA structure contains a narrow minor groove region on the 3' side of the core motif.

We re-thought the definition of our shape logos and agree with the Reviewer's concerns regarding our original concept. In response, we modified the shape logo definition (p. 15, l. 11-19, Figs 6C, D). We no longer use feature weights for generating DNA shape logos. Instead, we now use the relative R^2 gain in percentage as height of the letters in the logos. We agree that even with this update, there are still limits to the method. These limitations are addressed in detail in an expanded discussion. We do think that introducing the concept of shape logos will suggest many powerful applications of this study.

This work provides DNA-shape based binding models for more transcription factors from diverse families compared to previous work (Zhao et al, 2015). The most interesting part of the manuscript for me was that shape models achieve similar performance as 3mer based models with far fewer parameters. Overall, the first three claims are currently weakly supported. The last claim about mechanistic insight is not supported and should be moved to the discussion section.

We thank this Reviewer for appreciating important parts of the manuscript and we feel confident that we have now addressed the valid concerns raised and expanded the discussion of limitations where appropriate

Minor comment:

1. Legend for Figure S4 is incorrect.

The supplementary figures have been reorganized.

Reviewer #2:

Yang, et al. present a comprehensive analysis of DNA shape readout across a large set of mammalian transcription factors spanning several structural families. The work is based on resequencing (and thereby extending) 548 previously published HT-SELEX experiments, which provide a quantitative metric of in vitro TF-DNA binding affinity across a large number of DNA sequences. These data were used to train regression models that use positional sequence information alone or sequence plus DNA shape parameters to predict binding affinity. The importance of DNA shape readout across various TFs (and at various positions within the TF binding site) is mainly analyzed via the relative performance of regression models that are trained with different combinations of sequence and shape data. The work concludes that shape readout is important for DNA binding recognition across many TF families, although the relative positions affected by shape readout vary within and between TF structural families.

General comments:

Methodologically, this work does not substantially extend on the authors' previous publications. While the HT-SELEX resequencing has produced a more statistically robust data resource that will be of use to the broader community, this effort does not qualitatively change the nature of the information gained on the profiled TFs. Similarly, the computational methods in this work are straightforward applications of methods previously presented in Zhou, et al. 2015 and Abe, et al. 2015. That said, it is worth noting that the analyses presented in Fig 3 nicely reconcile a point of confusion in the Zhou, et al. manuscript, and now more firmly confirm that DNA shape parameters provide an efficient encoding of higher-order sequence information at binding sites. The greater significance of this work centers on its survey of DNA shape influences across a large number of TFs from various structural classes. As far as I am aware, this is the most comprehensive examination of the relative importance of DNA shape readout across diverse TFs to date.

We thank this Reviewer for both their frank opinion and appreciation for this study as indeed most comprehensive examination of the importance of DNA shape readout across a large number of TF families. We agree that this study more firmly confirms that DNA shape provides an efficient encoding of higher-order sequence information.

In the revision, we expanded the feature selection analysis to include testing of the individual contribution of each DNA shape feature category separately at each nucleotide position (p. 13, l. 7-9, Fig S5). The resulting heat maps provide a more detailed guide on the importance of each DNA shape feature at every position. These insights are all based on sequencing data without solving a single structure. To the best of our knowledge, this is done for the first time in the field and suggests a forward-looking approach to deriving structural information from sequencing data.

Major points

1) Given that the main novelty of this work is in the survey of DNA shape readout across TF families, it is disappointing that more interpretation and discussion is not focused on these results. For example, it may have been informative to move beyond homeodomains to discuss in more detail other TF families and how DNA shape influences their binding affinity (perhaps referencing representative protein-DNA structures for each family).

We share the concern raised by the Reviewer. In response to these concerns, we added a structure from the nuclear receptor family as an example of how the heat maps produced by our computational analyses of sequencing data can potentially reveal structural binding mechanisms (p. 15, l. 1-6, Fig S6). We did not achieve the same goal for each TF family due to a few reasons beyond the goals of this study. First, although the Protein Data Bank (PDB) has significantly expanded over the years, structure availability is still quite limited; most of the TFs in this study do not have a TF-DNA complex structure in the PDB. Second, we do not believe that individual structures are the ultimate answer, especially for those solved by X-ray crystallography, which as its name suggests, solves structures only in crystalized solid state instead of solution state. It is well documented that molecules in a crystal can suffer from deformations due to crystal packing effects.

2) One worrying aspect of the presented survey in Figs 5 & S6 was that the two feature selection strategies (adding shape info versus removing shape info at position i in the regression model)

provide several conflicting results. For example, the two strategies point to shape being important on opposite sides of the core motif for Lhx4 and others in Fig 5, and yield opposing conclusions on the positional influence of DNA shape for several families in Fig S6 (e.g. bHLH, bZIP, POU). How can these conflicts be explained/reconciled?

We agree that the two feature selection schemes provide conflicting results for some datasets. We now discuss this observation better and provide a partial solution to this limitation of the current data (p. 17-18, l. 21-23, 1-23). Both the red and blue heat maps aim to visualize the DNA shape importance at the individual positions. However, the red heat maps can contain false positives, and the blue heat maps can contain both false positives and false negatives. The DNA shape features at a position essentially reflect the pentamer context at that position. So shape features of adjacent positions may contain some redundant information. As a result, a position showing red may be due to the fact that DNA shape features at the position adjacent to it are important, thus resulting in false positives in the red heat map. On the other hand, for the same reason, a position may not show blue in the blue heat map due to the fact that its “neighbor” is compensating for it, causing false negatives in the heat maps. Moreover, the DNA shape features used here are ultimately derived from sequence. So a position showing blue may be due to the loss of this sequence information, resulting in false positives in the blue heat maps.

To facilitate the use of the heat maps given the aforementioned limitations, we generated an “overlay” purple version of the heat maps, in which each cell is the minimum of the red and blue heat maps (p. 13, l. 1-4, Figs 5C, S4). This takes away some noise. In addition, we expanded the feature selection analysis to further break down the DNA shape contribution into individual DNA shape feature categories, helping to locate the important shape features. In summary, the heat map analysis is a guide for locating the positions where DNA shape plays a role. Prominent positions in the “overlay” heat maps should provide us with the strongest confidence of the DNA shape effects. A particular scenario worth noting here is when the red heat map shows prominent shape effects in multiple consecutive positions, whereas at the same time, the blue heat map shows almost no effects. This we believe is due to false positives in the red and false negatives in the blue heat map. What we learn in this scenario is that DNA shape is important somewhere in the consecutive red regions, but we failed to locate it, even with the help of the blue heat map. These considerations are now addressed in the Discussion section.

3) I'm confused about how Fig 6 is claimed to support the conclusions derived from Fig 5. It may be helpful if the bases are labeled in Fig 6 A & B, but it seems that the narrowing MGW occurs at the center of the core motif, and not at the 3' side of the motif as expected from the heatmap in Fig 5. Similarly, and for the same reasons, I would have expected to see larger feature weights at positions 7 & 8 in the DNA shape logos presented in Fig 6 C & D.

We thank the Reviewer for sharing this confusion. To clarify the alignment, we now added Curves-derived plots of MGW showing the exact positions of MG narrowing (Fig 6).

4) It's unfortunate that only one protein (Max) has been characterized by both gcPBM and HT-SELEX, as it leaves only one dataset with which to assess the marginal value of deeper HT-SELEX coverage. Are there any alternate metrics with which to assess this point?

We share the Reviewer's concern for the sparse overlap between gcPBM and HT-SELEX data coverage but cannot change this situation in the context of this study. To see if lower coverage datasets could help, we processed the previously published HT-SELEX data using our pipeline, and only 22 proteins passed the filtering, compared to 218 with our deeper sequencing data. This is another demonstration of the benefit of deeper sequencing. We added this point to the revised manuscript (p. 7, l. 6-8).

5) I'm not sure how to interpret the PCA-based analyses. While the first two principal components in Fig 2B (1-mers + shape) account for more variance than those in Fig 2A (1-mers alone), are the two PCA projections truly comparable in this way? Is it possible that the greater amount of explained variance is due somehow to the differences in dataset dimensionality? The intra- & inter-family distance analyses based on the PCA plots are similarly difficult to interpret. It's claimed that the difference in median inter and intra-family group distances is larger when DNA shape features are incorporated in the model. However, that only seems true in absolute (as opposed to relative) terms.

This is a valid point raised by the Reviewer. To test our hypotheses, an additional control figure was generated by using randomly generated values as DNA shape features (p. 8, l. 4-8, Fig S2). In this control, the portion of variance explained dropped dramatically. Moreover, the inter- and intra-family group distance also dropped dramatically. This ruled out the possibility that the observation is simply due to higher dimensionality of the feature space.

Minor points

6) p.7 states that the regression model used is similar to the methodology in Zhou, et al. 2015. However, L2-regularized MLR is used in this work, whereas an SVR approach was used in Zhou, et al. While these methods perform similarly, it may be more correct to point to the Yang, et al. 2014 or Abe, et al. 2015 instead.

We thank the Reviewer for pointing this out. We cited Zhou, et al. 2015 to emphasize that second-order DNA shape features were also used in this study, which was not the case for the Yang, et al. 2014 paper. We now added also Yang, et al. 2014 as a relevant citation (p. 8, l. 14).

7) I find the new DNA shape logos quite informative. One problem with this representation is the lack of natural limits on the y-axes. Can any guidance be provided on that aspect?

We thank this Reviewer for their interest in our introduction of DNA shape logos. With our new definition of shape logos, the y-axis signifies the gain in R^2 performance (p. 15, l. 11-19, Figs 6C, D).

Reviewer #3:

Yang, et al. describe a large-scale computational analysis to uncover the contribution of DNA shape to transcription factor (TF) binding. To carry out this analysis, they use HT-SELEX data describing the binding preferences of hundreds of mammalian TFs and augment this data set with increased sequencing depth. They use models that describe TF binding preferences with and without shape parameters to show that shape is important for many families of TFs and use two approaches to identify specific positions at which shape is important. Finally, they introduce a sequence logo to display TF shape preferences.

Over the last few years, the importance of DNA shape in TF binding has been highlighted by several papers, including many by Rohs, one of the corresponding authors of this paper. The novelty of this paper comes from two features: (1) the authors perform an analysis of the role of DNA shape on a larger set of TFs, representing more DNA binding domain families, than previous studies (e.g. Zhou, et al. 2015 PNAS), and (2) they measure base pair-specific information about the contribution of DNA shape to TF binding and encode this information in a DNA shape logo. This work should be of interest to large community of scientists interested in understanding TF-DNA interactions. The conclusions in the paper are generally well-supported by the data, but some parts of the text are difficult to understand, making some claims harder to evaluate. The figures are generally clear and easy to interpret. The authors did a nice job of providing a comprehensive amount of supplementary data, including the details of all the data sets used, the models for each TF, etc.

We thank this Reviewer for summarizing the key advances brought forward in this comprehensive study and for evaluating the conclusions as well supported. We agree with the Reviewer that some sections of the text were difficult to understand and, in response, edited the text and removed multiple unnecessary sentences.

Major suggestions:

- For clarity, it would be useful to start paragraph 2 with the goal of selecting the core motif. I got the idea after reading the whole paragraph, but it took a little while. In general, the paper could be vastly improved by a thorough editing of the text. In particular, starting paragraphs with topic sentences that explain either the rationale or goal for the next step of analysis or the connection to the previous step would make the paper far more readable.

We thank this Reviewer for their frank advice on text editing. We have edited the text throughout. We added the goal of choosing a core motif in the first sentence of paragraph 2 (p. 5-6, l. 19-23, 1-2). We added topic sentences to several paragraphs to clarify the goal of the analysis (e.g., p. 6, l. 9).

- What is the rationale for the formula for choosing the number of allowed mismatches? Do the results depend on the details of this formula?

The rationale is to allow variability in potential binding sites while filtering out oligos that are unlikely to bound. This is now explained in detail in the Materials and Methods section (p. 24, l. 6-23).

In the revised manuscript, we used a relaxed threshold for the core motif to increase dataset variability. We compared the performance to a more stringent threshold (p. 10, l. 1-3, Fig S3D). The added variability improved binding prediction marginally, and the high correlation between them indicates the robustness of the method on the choice of thresholds ($r=0.85$).

- What is the rationale for the formula for choosing the initial and maximum length of the flanking regions? Again, do the results depend strongly on the details of this formula?

In principle we wish to make the flanking regions (and hence M) as large as possible. However, the number of possible M -words grows exponentially with M , while the total sample size remains constant (in fact the number of M -words slightly decreases with M). Thus, the average count per M -word decreases exponentially with M , and signal to noise ratio drops. Because of this tradeoff, we prefer to try several values of M . Shape prediction at a particular position requires statistics on 5-mers centered at that word. By setting the minimum value of M to 10 we guarantee to get shape information for at least 6 positions. If possible, we increase M depending on our sample: we pick the largest M for which the top M -word has counts ≥ 100 and at least 1,000 M -words have reliable counts (i.e. count > 8), in order to have enough variability in the data to learn a model. This is now explained in detail in the Materials and Methods section (p. 23-24, l. 12-23, 1-5).

In the revised manuscript, we compared the performance achieved by 10-words to 12-words (p. 10, l. 1-3, Fig S3E). Performance using 10-words was higher, since their binding scores are more accurate due to higher counts. However, the correlation between 10-words and 12-words results was quite high ($r=0.64$). In our analyses, we tested different M 's for each TF, and selected the value with the highest R^2 .

- It's not clear to me how "good" of a correlation 0.64 is and how the error in the M -word scores will influence the conclusions. It is possible to validate M -word scores for any other proteins using regular PBMs?

We understand this concern of the Reviewer. Unfortunately, the only available data for other proteins for which we can deduce longer M -word scores is genomic-context PBMs (gcPBMs). Universal PBMs (uPBMs), for which data is available for more than 1,000 proteins, can only give robust binding scores for 8-words, as each 10-mer appears only once. Thus, we could only compare to the one protein in the overlap between HT-SELEX and gcPBM data.

To gauge how good the correlation 0.64 is, we calculated the M -word correlation of HT-SELEX replicate experiments using $M=12$. The mean Pearson correlation of HT-SELEX replicates was 0.68 and std 0.28. Hence, 0.64 is at about the same level as correlation between replicates.

- In looking at the shape importance figures in S6, I noticed that between the two feature selection protocols, some blue and red heat maps seemed correlated (e.g. the RFX proteins) and some seemed anti-correlated (e.g. TFAP). What is the significance of this observation? Or in Figure 5, some proteins, e.g. Lhx8 show a correlation between the blue and red heat maps and others, e.g. Lhx4, show an anti-correlation. It would be nice to see a deeper discussion of the significance of the red and blue heat maps when compared to each other.

We agree that the two feature selection schemes provide conflicting results for some datasets, which is also pointed out by reviewer #2. We now discuss this observation and provide a partial solution (p. 17-18, l. 21-23, 1-23). In summary, The DNA shape features at a position essentially reflect the

pentamer context at that position. So shape features of adjacent positions may contain redundant information. As a result, a position showing red may be due to the fact that DNA shape features at the position adjacent to it are important, thus resulting in false positives in the red heat map. On the other hand, for the same reason, a position may not show blue in the blue heat map due to the fact that its "neighbor" is making up for it, causing false negatives in the heat maps. A particular scenario is when the red heat map shows prominent shape effects in multiple consecutive positions, whereas at the same time, the blue heat map shows almost no effects. This we believe is due to false positives in the red and false negatives in the blue heat map. This indicates that DNA shape is important somewhere in the consecutive red regions, but we failed to locate the precise position, even with the help of the blue heat map.

Minor suggestions:

- The first 3 sentences of the introduction are rather convoluted and could be simplified for clarity.

We agree and revised the first 3 sentences to make them clearer (p. 3, l. 3-5).

- It would be helpful to include a sentence in the introduction about the TFs included in the HT-SELEX database. Which mammals are they from? How were they selected?

The experiments cover mouse and human proteins selected to span 40 TF families. We now added this information to the manuscript (p. 5, l. 8-9).

- In paragraph 1 of results, a total of 197 TFs were said to pass quality control, but in Figure 1, 218 seem to pass the first round of QC.

The third box in Figure 1 states that 218 datasets passed the filtering, but the last box states 197 passed the final data filtering based on the R^2 results. We now revised the last sentence of the paragraph to clarify these two steps (p. 5, l. 16-18). (Note that the numbers have changed in the revision due to our choice to relax the mismatch threshold for identifying binding sites).

- page 5, line 7 "Increased sequencing depth will allow us" - most of the paper is written in the past tense. I would keep it consistent here.

We agree and corrected it to past tense (p. 5, l. 10).

- The description of the algorithm to determine maximum length of flanking regions was really hard to understand. (E.g. in the example given, I'm not exactly sure how "3" is the answer.)

We revised the description of the algorithm (p. 23-24, l. 12-23, 1-5).

- I would highlight more prominently the result in Figure S1C - you tried alternative M-word scoring, which didn't perform as well.

We added that we tested both the use of the new data and the different binding scores, and that the ratio to estimated frequencies performed best (p. 7, l. 4-6).

- In the results (page 7), and/or in a figure, it would be useful to enumerate the DNA shape features that are encoded into the feature vectors.

The DNA shape features are now enumerated when mentioned for the first time in the results section (p. 7, l. 16-18).

- I think the Methods section could be re-ordered to match the order that each method is used in the results section more closely.

The Methods section is now ordered as the following:

1. Description of the preprocessing of the data
2. Description of the PCA and regression methods
3. Description of the method for generating shape logos

It reflects the order of the results in the manuscript. We appreciate this suggestion.

- It would be helpful to have a table with each row corresponding to a TF family, and columns for the number of representatives, fraction that are improved by including shape, average difference between r^2 with and without shape. (This might go in the main text.)

Supplementary Table EV1 now contains such information for each TF.

- I didn't understand the claim on page 9 about homeodomains binding as monomers. Was the goal to justify why you would see shape preferences for homeodomains, even as monomers? Or something else? Please clarify.

Previously, the importance of DNA shape among homeodomain TFs was demonstrated mainly for cooperatively binding homeodomain TFs, or Hox proteins with co-factors, and DNA shape recognition might be due to degenerate positions in the DNA spacer separating the two homeodomains when they bind cooperatively (Slattery et al. *Cell* 2011, Abe et al. *Cell*, 2015). Here in our study, we focus only on the condition in which the TFs bind as monomers to demonstrate the general importance of DNA shape. It is very likely that the role of shape readout generally increases when we study cooperative binding of TFs in general, homeodomains included.

- Can you include the same details in the text for how you calculated the % change in R^2 for Figure 5B as you did for Figure 5A?

We thank the Reviewer for the suggestion. We have added the formula (p. 12-13, l. 23, 1).

- In general, the figure legends could be expanded to help the reader understand what to take away from each figure. Many of the supplementary figures didn't have any legends, and for some, I really needed them (e.g. Figure S2).

We thank the Reviewer for pointing this out. Supplementary Figures have now been reorganized, and the main text has been updated to better explain the figures. In addition, the figure captions have been expanded to better describe the message conveyed in each figure.

2nd Editorial Decision

20 December 2016

Thank you again for sending us your revised manuscript. We have now heard back from the two reviewers who were asked to evaluate your revised study. As you will see below, the reviewers think that the manuscript is improved and most issues have been satisfactorily addressed. However, reviewer #1 refers to the need to perform some text modifications, in order to avoid overstatements. As such, we would ask you to perform these changes in a minor revision.

REFeree REPORTS

Reviewer #1:

The revised manuscript from Yang and coworkers provides evidence that incorporating DNA shape features into models of transcription factor-binding improves these models for TFs from different families. The study also provides a good TF-DNA binding data resource. The claims are generally supported by the data. The paper might be improved by qualifying the claims to better reflect the current predictive power of shape features.

With respect to the specific claims of the manuscript:

1) DNA shape contribution to TF-DNA binding preferences is prevalent across many TF families

The addition of randomized shape models adds strong support to this claim. It is remarkable that shape features improve TF-binding models while adding so many fewer parameters compared to 2-

mer/3-mer features. However, the extent to which shape contributes to TF-binding could be discussed in a more balanced manner. While the evidence does support the claim that shape effects are "prevalent", these shape effects are quite small in most cases. Currently the manuscript might be misinterpreted as claiming that shape is the predominant predictor of TF-binding. This is a concern especially because most of the signal for inter-family distance comes from the separation of the homeodomain family from other families (Figure 2).

- 2) DNA shape features in core-motif flanks are important for many TF families
- 3) DNA shape contribution has TF-family-specific positional preferences within the binding site

The data do support both claims, but that support is also weak because of the false positives and false negatives in the heat maps. I also note that almost all of the data supporting these two claims are in supplemental files (The 3' effect shown in Figure 5 is still in the core motif, not the flank). It also might be helpful if the authors could comment on how future studies might decrease the rates of false positive and false negative shape features, rather than just acknowledging that false positives and false negatives exist. Are we stuck forever with high error rates or can they be corrected?

- 4) DNA shape features provide mechanistic insights into TF-DNA binding.

The authors have softened this claim by pointing out that the shape logos do not lead to a prediction of a protein-DNA structure model, but are a general guide for DNA shape preferences. However the authors do say that their data provide insight in the absence of all atom simulations. It may be useful to draw a distinction here between an "observation" (e.g that narrow minor grooves are preferred 3' of some homeodomains) and an "insight" (that homeodomains slide a helix into the minor groove).

Overall, the primary contribution of this study is to provide improved models of TF-binding preferences for TFs across various families by incorporating shape features that capture dependencies between nucleotides.

Reviewer #2:

The revised manuscript appropriately addresses most of my previous comments. In particular, the revised text more clearly discusses the potential limitations of the two feature selection schemes used for assessing position-specific shape dependencies. The new purple heatmaps also provide a simple and conservative integration of the information gained via the two feature selection schemes. The assessment of DNA shape dependencies across all examined TF families is also improved by the new Fig S5, which breaks out analyses according to each of the four DNA shape parameters.

My overall opinion of the manuscript remains that the computational methodology and analytical approach do not substantially extend on the authors' previous work, and many of the results confirm and improve on older results rather than adding something completely novel (e.g. Fig 3). However, the current work represents an excellent summarization of DNA-shape dependencies across a broad range of transcription factors. As such, I can see this manuscript being a highly useful reference that many in the field return to over the coming years.

2nd Revision - authors' response

04 January 2017

Reviewer #1:

The revised manuscript from Yang and coworkers provides evidence that incorporating DNA shape features into models of transcription factor-binding improves these models for TFs from different families. The study also provides a good TF-DNA binding data resource. The claims are generally supported by the data. The paper might be improved by qualifying the claims to better reflect the current predictive power of shape features.

We thank Reviewer 1 for stating that the claims in our manuscript are generally supported by the data. The point is also well taken that the text might be improved by qualifying the claims to better reflect the actual predictive power of shape features.

To address this valid concern, we edited the manuscript throughout in order to ensure that there is no overstatement of our claims. We particularly tried to clarify that DNA shape alone is of course insufficient to explain TF binding, as discussed in all of our previous work, and emphasized that instead DNA shape represents an additional layer to sequence that contributes to TF binding.

We made changes to clarify this important point on page 8 in line 23 and on page 9 in line 1 by making the red edits in the following sentence: “With DNA sequence readout playing a dominant role in TF binding, the importance of DNA shape recognition as additional contribution varied both between and within TF families.”

For the same reason, we added the following text on page 18 in line 4, here in red font: “First, DNA shape alone is obviously insufficient to explain TF binding (Zhou et al, 2015).”

Accordingly, we further clarified on page 19 in lines 18-20: “In conclusion, while DNA sequence describes opportunities to form hydrogen bonds and other direct contacts between amino acids and bases, DNA shape can provide an important additional contribution to TF binding (Rohs et al, 2010; Rohs et al, 2009).”

We made several additional smaller changes in all sections of the text to prevent any potential overstatement or, maybe more importantly, any possible misunderstanding by readers of the paper. This is absolutely in our interest and we thank the Reviewer for pointing this out.

With respect to the specific claims of the manuscript:

1) DNA shape contribution to TF-DNA binding preferences is prevalent across many TF families

The addition of randomized shape models adds strong support to this claim. It is remarkable that shape features improve TF-binding models while adding so many fewer parameters compared to 2-mer/3-mer features. However, the extent to which shape contributes to TF-binding could be discussed in a more balanced manner. While the evidence does support the claim that shape effects are "prevalent", these shape effects are quite small in most cases. Currently the manuscript might be misinterpreted as claiming that shape is the predominant predictor of TF-binding. This is a concern especially because most of the signal for inter-family distance comes from the separation of the homeodomain family from other families (Figure 2).

We agree with Reviewer 1 that the comparison of models using DNashape-derived structural features vs. randomized structural features is an important result. We therefore moved the panel showing these results from the Appendix (previously Fig S3B) to the main manuscript (now Fig 3B).

As discussed above, we edited the text throughout to ensure a balanced discussion of shape effects. The text did previously state on page 9 in lines 8-9: “At least half of the members in each of these families, covered by our data, showed greater than 10% performance improvement when DNA shape features were added to the model.” We verified the text throughout to indicate that DNA shape is an additional contribution and not the dominant effect. To clarify that most of the inter-family distance comes from the separation of the homeodomain family, we now include an additional statement on page 8 in lines 4-5: “... indicating that more variance could be explained by introducing DNA shape features, in part due to the better separation of the homeodomain family (Fig 2B).”

2) DNA shape features in core-motif flanks are important for many TF families

3) DNA shape contribution has TF-family-specific positional preferences within the binding site

The data do support both claims, but that support is also weak because of the false positives and false negatives in the heat maps. I also note that almost all of the data supporting these two claims are in supplemental files (The 3' effect shown in Figure 5 is still in the core motif, not the flank). It

also might be helpful if the authors could comment on how future studies might decrease the rates of false positive and false negative shape features, rather than just acknowledging that false positives and false negatives exist. Are we stuck forever with high error rates or can they be corrected?

We agree with Reviewer 1 also here and think that the limitations of the heat map analysis are discussed and presented in a balanced way. We moved one of the figures evaluating flanking effects from the Appendix (previously Fig S3F) to the main manuscript (now Fig 3E). This panel directly addresses the importance of shape information at the edges flanking the core motifs.

We also edited and improved a statement on future studies on page 19 in lines 8-10 with modified text in red font: “in the future, such heat map analysis, when combined with TF-DNA binding measurements of improved quality, will allow us to gain more clues of TF-DNA binding mechanisms from DNA sequencing data.”

4) DNA shape features provide mechanistic insights into TF-DNA binding.

The authors have softened this claim by pointing out that the shape logos do not lead to a prediction of a protein-DNA structure model, but are a general guide for DNA shape preferences. However the authors do say that their data provide insight in the absence of all atom simulations. It may be useful to draw a distinction here between an "observation" (e.g that narrow minor grooves are preferred 3' of some homeodomains) and an "insight" (that homeodomains slide a helix into the minor groove).

This point is also well taken and partially addressed by additional statements discussed above. We changed “insights” to “clues” in several instances. We modified a sentence on page 10 in lines 17-18 as follows: “Here, we gained additional clues for possible explanations of this observation.” We also replaced “insights” by “clues” on page 19 in line 9.

Overall, the primary contribution of this study is to provide improved models of TF-binding preferences for TFs across various families by incorporating shape features that capture dependencies between nucleotides.

We thank Reviewer 1 for their very insightful and constructive suggestions, which have improved the presentation of the results.

Reviewer #2:

The revised manuscript appropriately addresses most of my previous comments. In particular, the revised text more clearly discusses the potential limitations of the two feature selection schemes used for assessing position-specific shape dependencies. The new purple heatmaps also provide a simple and conservative integration of the information gained via the two feature selection schemes. The assessment of DNA shape dependencies across all examined TF families is also improved by the new Fig S5, which breaks out analyses according to each of the four DNA shape parameters.

My overall opinion of the manuscript remains that the computational methodology and analytical approach do not substantially extend on the authors' previous work, and many of the results confirm and improve on older results rather than adding something completely novel (e.g. Fig 3). However, the current work represents an excellent summarization of DNA-shape dependencies across a broad range of transcription factors. As such, I can see this manuscript being a highly useful reference that many in the field return to over the coming years.

We thank Reviewer 2 for their very helpful suggestions, which have definitely improved the manuscript and data presentation.

3rd Editorial Decision

05 January 2017

Thank you for sending us your revised manuscript. We are now satisfied with the modifications made and I am pleased to inform you that your paper has been accepted for publication.

Corresponding Author Name: Remo Rohs

Manuscript Number: MSB-16-7238